# Report

# A protein-based classifier for differentiating follicular thyroid adenoma and carcinoma

Yaoting Sun [iD] [1,2,3,41], He Wang [iD] [1,2,3,41], Lu Li [1,2,3,4,41], Jianbiao Wang [5,41], Wanyuan Chen [6,41], Li Peng [iD] [7,41], Pingping Hu [1,2,3], Jing Yu [1,2,3], Xue Cai [1,2,3], Nan Yao [1,2,3], Yan Zhou [1,2,3], Jiatong Wang [1,2,3], Yingrui Wang [iD] [1,2,3], Liqin Qian [1,2,3], Weigang Ge [iD] [8], Mengni Chen [iD] [8], Feng Yang [9], Zhiqiang Gui [10], Wei Sun [10], Zhihong Wang [10], Minghua Ge [11], Yi He [12], Guangzhi Wang [13], Yongfu Zhao [13], Huanjie Chen [14], Xiaohong Wu [15], Yuxin Du [16], Wenjun Wei [16], Fan Wu [17], Dingcun Luo [17], Xiangfeng Lin [18], Haitao Zheng [18], Xin Zhu [19], Bei Wei [19,20], Jiafei Shen [19,20], Jincao Yao [19,20], Zhennan Yuan [21], Tong Liu [22], Jun Pan [23], Yifeng Zhang [24], Yangfan Lv [25], Qiaonan Guo [25], Qijun Wu [26], Tingting Gong [27], Ting Chen [28], Shu Zheng [28], Jingqiang Zhu [29], Hanqing Liu [30], Chuang Chen [iD] [30], Hong Han [31], Sathiyamoorthy Selvarajan [32], Michael Mingzhao Xing [33], Kennichi Kakudo [iD] [34], Erik K Alexander [35], Yijun Wu [23], Yu Wang [16], Dong Xu [19,20], Hao Zhang [10], Xiu Nie [7], Oi Lian Kon [36], N Gopalakrishna Iyer [37], Zhiyan Liu [iD] [38], Yi Zhu [iD] [1,2,3✉], Haixia Guan [iD] [39✉], Tiannan Guo [iD] [1,2,3,40✉] & We-TEC Investigators*

## Abstract

Differentiating follicular thyroid adenoma (FTA) from carcinoma (FTC) remains challenging due to similar histological features separate from invasion. This study developed and validated DNA- and/or protein-based classifiers. A total of 2443 thyroid samples from 1568 patients were obtained from 24 centers in China and Singapore. Next-generation sequencing of a 66-gene panel revealed 41 (62.1%) detectable genes, while 25 were not, showing similar alteration patterns with differing mutation frequencies. Proteomics quantified 10,336 proteins, with 187 dysregulated. A discovery protein-based XGBoost model achieved an AUROC of 0.899 (95% CI, 0.849–0.949), outperforming the gene-based model (AUROC 0.670 [95% CI, 0.612–0.729]). A subsequent 24-protein classifier, developed via targeted mass spectrometry and validated in three independent sets, showed high performance in retrospective cohorts (AUROC 0.871 [95% CI, 0.833–0.910] and 0.853 [95% CI, 0.772–0.934]) and prospective biopsies (AUROC 0.781 [95% CI, 0.563–1.000]). It exhibited a 95.7% negative predictive value for ruling out malignancy. This study presents a promising protein-based approach for the differential diagnosis of FTA and FTC, potentially enhancing diagnostic accuracy and clinical decision-making.

**Keywords** Follicular Thyroid Adenoma; Follicular Thyroid Carcinoma; Proteomics; Gene Mutation; Machine Learning
**Subject Categories** Biomarkers; Cancer; Proteomics

## Introduction

The incidence of thyroid nodules and thyroid cancer has continuously increased over the past decades (Boucai et al, 2024). Although ultrasonography and ultrasound-guided fine needle aspiration (FNA) improve the distinction between benign and malignant nodules, approximately 10–30% of nodules still cannot be definitively diagnosed by FNA and require surgical diagnosis (Fagin and Wells, 2016). These indeterminate thyroid nodules (ITN) are mainly composed of follicular thyroid adenoma (FTA) and follicular thyroid carcinoma (FTC)(Alexander et al, 2012).

The incidence of benign FTA is approximately five times higher than that of invasive FTC in surgical specimens (McHenry and Phitayakorn, 2011). Among all FTC patients, 7–23% will have distant metastases at diagnosis or during follow-up (Daniels, 2018; Wu et al, 2022), and 11–39% will experience recurrence (Jung et al, 2016; Zhang et al, 2023). The 10-year disease-specific mortality rate for invasive FTC is 15–28%. Therefore, it is crucial to accurately differentiate FTC from benign conditions to ensure appropriate clinical management and prognosis.

The differential diagnosis between FTA and FTC is one of the most challenging aspects of thyroid pathology due to their subtle differences. Currently, the standard for stratification is based on capsular and vascular invasion (Gromova et al, 2023). FTA and

A full list of author affiliations appears at the end of the paper. *A list of authors and their affiliations appears at the end of the paper. ✉E-mail: zhuyi@westlake.edu.cn; guanhaixia@gdph.org.cn; guotiannan@westlake.edu.cn

FTC cannot be distinguished preoperatively because capsular invasion cannot be assessed by cytology, ultrasound, or clinical features. The only way to differentiate them is through diagnostic surgery followed by histopathological examination by experienced pathologists. However, even in postoperative histopathology, FTC and FTA can be challenging to distinguish as FTC often closely resembles FTA microscopically. Capsular invasion must be carefully inspected under the microscope with serial sections to make a judgment. Sometimes, due to insufficient sampling, pathologists are unable to examine the entire capsule in the tissue specimen, making it difficult to provide a definitive diagnosis.

Nucleic acid-based molecular testing has been developed and validated for assistance with ITN and has achieved high negative predictive values (>95%) (Alexander et al, 2012; Livhits et al, 2021; Patel et al, 2018). However, genomic and transcriptomic signatures distinguishing FTA from FTC have not yet been identified. *RAS* mutations and PAX8/PPARγ rearrangements are common alterations in follicular neoplasms but can be detected in both, so individual gene alterations cannot distinguish the two. Proteomics provides phenotypic validation and interpretation of genomics, enabling more precise and reliable information for early detection of liver disease (Niu et al, 2022) and cardiovascular events (Helgason et al, 2023), classification of benign/malignant thyroid nodules (Shi et al, 2022), personalized prognostication (Wang et al, 2024), and molecular characterization of specialized histopathological conditions (Huang et al, 2022; Li et al, 2024b). Thus, proteomics drives the development of precision medicine. The above examples illustrate the promising prospects of proteomics technologies in the discovery of clinical biomarkers and drug targets, as well as disease classification.

In this study, we employed deep discovery and targeted proteomic strategies to select a panel of proteins and built a classifier for stratifying follicular neoplasms in retrospective tissue samples. The model was further validated in two independent retrospective tissue sets and one prospective biopsy set.

# Results

## Patient characteristics and study design

In the present study, we collected a total of 2443 data files which were analyzed for gene alterations and proteome profiling, from 1568 patients (including 909 with FTA and 659 with FTC) across 24 centers in China and Singapore. The median age at diagnosis was 49.0 years (interquartile range [IQR, Q1-Q3]: 36.0–60.0). There were 1105 females and 463 males, with a female-to-male ratio of 2.4:1. The median nodule size was 35 mm, with an IQR of 25.0 to 48.0 mm. Nodule sizes less than 40 mm were observed in 893 cases (57.0%), while sizes greater than or equal to 40 mm were observed in 667 cases (42.5%). Detailed patient information is listed in Table 1.

To profile the molecular landscape of follicular tumors, we first analyzed gene alterations of 609 samples from 609 patients by next-generation sequencing (NGS) and proteome differences between FTC and FTA of 620 samples from 614 patients by tandem mass tag (TMT)-based discovery proteomics. Next, we performed targeted proteomics through parallel reaction monitoring (PRM) on 729 samples to construct a classifier for stratifying FTC and FTA. The classifier was further validated in three testing sets including an internal testing

set ($n = 325$) and two independent retrospective ($n = 90$) and prospective ($n = 70$) testing sets (Fig. 1). Finally, we compared the performance of combined gene panel data and targeted proteomics data from the same 494 patients against individual feature types to assess improved differentiation capability.

## The gene-panel-based model cannot precisely distinguish FTA and FTC

We initially screened a 66-gene panel across 609 samples, including 396 FTA and 213 FTC. Among the 66 target genes, 41 genes (62.1%) were successfully detected in the present dataset, while the remaining genes were not. The gene analysis revealed that 325 of the 609 samples (53.4%) carried mutations, specifically 183/396 (46.2%) in FA and 142/213 (66.7%) in FTC. Moreover, 91 of 609 samples carried at least two mutations. The above data indicate that DNA alterations are uninformative of the histological diagnosis of 46.6% of follicular neoplasms.

The results exhibited a similar gene alteration pattern with different mutant frequency between FTA and FTC. The five most common mutations were *NRAS* (12.4% vs. 21.6%), *HRAS* (7.3% vs. 14.1%), *TERT* (2.3% vs. 18.8%), *DICER1* (4.0% vs. 10.3%), and *EIF1AX* (5.8% vs. 6.6%), with the numbers in parentheses indicating the mutation frequencies in FTA and FTC, respectively (Appendix Table S1).

Subsequently, we constructed a gene-panel-based XGBoost model utilizing a matrix comprising 609 samples and 41 detected genes. The model was trained on a dataset comprising 296 samples from one center and subsequently tested on 313 samples from five independent centers. The model with four gene mutations (*TERT* promoter, *NRAS*, *DICER1* and *BRAF*) achieved an AUC of 0.670 (95% confidence interval [CI], 0.612–0.729), indicating that it was not sufficiently robust in the classification of FTA and FTC (Figs. 2A and EV1).

These findings suggest that gene mutations alone are insufficient to reliably distinguish between FTA and FTC which may be due to low-frequency gene alterations and their overlapping mutation patterns.

## In-depth proteomics analysis holds promise for differentiating FTA and FTC

Considering the similarities in biological morphology and gene expression between FTA and FTC, we employed an in-depth proteomics identification approach to analyze an FFPE dataset comprising 645 FFPE samples. From these samples, we quantified 10,336 proteins with a false discovery rate (FDR) of less than 1% at both peptide and protein levels. After quality control analysis (Appendix Fig. S1), we derived a matrix containing 7876 proteins from 620 samples (331 FTA and 289 FTC). The samples were further divided into a discovery set (485 samples, 261 FTA and 224 FTC) for protein feature selection and model construction, and an independent testing set (135 samples, 70 FTA and 65 FTC) for model performance evaluation.

Protein biomarker selection was conducted by comparison of FTC vs. FTA based on the discovery set which revealed 187 differentially expressed proteins (DEPs, Fig. EV2A). Enrichment analysis showed these proteins to be involved in functions and pathways of thyroid hormone generation and metabolic processes (Fig. EV2B). In

**Table 1.** Baseline characteristics of patients from different samples.

| Molecules | DNA | | |
|---|---|---|---|
| Detection | Next-generation sequencing (NGS) | | |
| Group | Training | Testing | All |
| Collection | Retrospective | Retrospective | |
| Sample type | FFPE | FFPE | |
| Total no. | | | |
| Centers | 1 | 5 | 6 |
| Patients | 296 | 313 | 609 |
| Data files[a] | 296 | 313 | 609 |
| Histopathology diagnosis | | | |
| FTA | 191 (64.5%) | 205 (65.5%) | 396 (65.0%) |
| FTC | 105 (35.5%) | 108 (34.5%) | 213 (35.0%) |
| Age at diagnosis | | | |
| Median | 45 | 52 | 48 |
| IQR (Q3-Q1) | 55.2–35.0 | 61.0–38.0 | 59.0–36.0 |
| <55 | 215 (72.6%) | 181 (57.8%) | 396 (65.0%) |
| ≥55 | 81 (27.4%) | 132 (42.2%) | 213 (35.0%) |
| Gender | | | |
| Female | 188 (63.5%) | 228 (72.8%) | 416 (68.3%) |
| Male | 108 (36.5%) | 85 (27.2%) | 193 (31.7%) |
| Nodule size[b] | | | |
| Median | 39.5 | 35 | 38 |
| IQR (Q3-Q1) | 50.0–28.0 | 47.0–25.0 | 49.0–27.0 |
| <40 | 147 (49.7%) | 182 (58.1%) | 329 (54.0%) |
| ≥40 | 147 (49.7%) | 131 (41.9%) | 278 (45.6%) |
| Molecules | Protein | | |
| Detection | Tandem mass tag-mass spectrometry (TMT-MS) | | |
| Group | Training | Testing | All |
| Collection | Retrospective | Retrospective | |
| Sample type | FFPE | FFPE | |
| Total no. | | | |
| Centers | 12 | 4 | 12 |
| Patients | 480 | 134 | 614 |
| Data files[a] | 485 | 135 | 620 |
| Histopathology diagnosis | | | |
| FTA | 260 (54.2%) | 69 (51.5%) | 329 (53.6%) |
| FTC | 220 (45.8%) | 65 (48.5%) | 285 (46.4%) |
| Age at diagnosis | | | |
| Median | 51 | 48 | 51 |
| IQR (Q3-Q1) | 62.0–38.0 | 58.0–34.0 | 61.0–37.0 |
| <55 | 275 (57.3%) | 90 (67.2%) | 365 (59.4%) |
| ≥55 | 205 (42.7%) | 44 (32.8%) | 249 (40.6%) |
| Gender | | | |
| Female | 349 (72.7%) | 96 (71.6%) | 445 (72.5%) |
| Male | 131 (27.3%) | 38 (28.4%) | 169 (27.5%) |

**Table 1.** (continued)

| Molecules | Protein | | |
| --- | --- | --- | --- |
| Detection | Tandem mass tag-mass spectrometry (TMT-MS) | | |
| Group | Training | Testing | All |
| Collection | Retrospective | Retrospective | |
| Sample type | FFPE | FFPE | |
| Nodule size[b] | | | |
| Median | 36 | 37 | 36 |
| IQR (Q3-Q1) | 50.0–25.0 | 49.8–26.0 | 50.0–25.0 |
| <40 | 263 (54.8%) | 71 (53.0%) | 334 (54.4%) |
| ≥40 | 213 (44.4%) | 63 (47.0%) | 276 (45.0%) |

| Molecules | Protein | | | | |
| --- | --- | --- | --- | --- | --- |
| Detection | Parallel reaction monitoring-mass spectrometry (PRM-MS) | | | | |
| Group | Training | Testing #1 | Testing #2 | Testing #3 | All |
| Collection | Retrospective | Retrospective | Retrospective | Prospective | |
| Sample type | FFPE | FFPE | FFPE | FNA | |
| Total no. | | | | | |
| Centers | 18 | 17 | 3 | 8 | 21 |
| Patients | 679 | 325 | 90 | 70 | 1164 |
| Data files[a] | 729 | 325 | 90 | 70 | 1214 |
| Histopathology diagnosis | | | | | |
| FTA | 389 (57.3%) | 201 (61.8%) | 37 (41.1%) | 59 (84.3%) | 686 (58.9%) |
| FTC | 290 (42.7%) | 124 (38.2%) | 53 (58.9%) | 11 (15.7%) | 478 (41.1%) |
| Age at diagnosis | | | | | |
| Median | 47 | 48 | 57.5 | 51 | 48 |
| IQR (Q3-Q1) | 57.0–35.0 | 57.0–35.0 | 66.0–41.2 | 60.0–40.2 | 58.0–36.0 |
| <55 | 469 (69.1%) | 217 (66.8%) | 37 (41.1%) | 41 (58.6%) | 764 (65.6%) |
| ≥55 | 210 (30.9%) | 108 (33.2%) | 53 (58.9%) | 29 (41.4%) | 400 (34.4%) |
| Gender | | | | | |
| Female | 475 (70.0%) | 217 (66.8%) | 67 (74.4%) | 56 (80.0%) | 815 (70.0%) |
| Male | 204 (30.0%) | 108 (33.2%) | 23 (25.6%) | 14 (20.0%) | 349 (30.0%) |
| Nodule size[b] | | | | | |
| Median | 35 | 35 | 37.65 | 29.95 | 35 |
| IQR (Q3-Q1) | 46.0–25.0 | 45.0–25.0 | 45.0–27.0 | 40.2–18.5 | 45.0–25.0 |
| <40 | 403 (59.4%) | 192 (59.1%) | 48 (53.3%) | 49 (70.0%) | 692 (59.5%) |
| ≥40 | 273 (40.2%) | 132 (40.6%) | 42 (46.7%) | 21 (30.0%) | 468 (40.2%) |

| Molecules | DNA+Protein | | | |
| --- | --- | --- | --- | --- |
| Detection | NGS + PRM | | | |
| Group | Training | Testing | All | All |
| Collection | Retrospective | Retrospective | | |
| Sample type | FFPE | FFPE | | |
| Total no. | | | | |
| Centers | 6 | 6 | 6 | 24 |
| Patients | 394 | 100 | 494 | 1568 |
| Data files[a] | 394 | 100 | 494 | 2443 |

**Table 1.** (continued)

| Molecules | DNA+Protein | | | All |
| --- | --- | --- | --- | --- |
| Detection | NGS + PRM | | | |
| Group | Training | Testing | All | |
| Collection | Retrospective | Retrospective | | |
| Sample type | FFPE | FFPE | | |
| **Histopathology diagnosis** | | | | |
| FTA | 266 (67.5%) | 58 (58.0%) | 324 (65.6%) | 909 (58.0%) |
| FTC | 128 (32.5%) | 42 (42.0%) | 170 (34.4%) | 659 (42.0%) |
| **Age at diagnosis** | | | | |
| Median | 52.5 | 47.5 | 49 | 49 |
| IQR (Q3-Q1) | 56.0–35.0 | 62.0–41.0 | 57.8–36.0 | 60.0–36.0 |
| <55 | 273 (69.3%) | 57 (57.0%) | 330 (66.8%) | 986 (62.9%) |
| ≥55 | 121 (30.7%) | 43 (43.0%) | 164 (33.2%) | 582 (37.1%) |
| **Gender** | | | | |
| Female | 258 (65.5%) | 75 (75.0%) | 333 (67.4%) | 1105 (70.5%) |
| Male | 136 (34.5%) | 25 (25.0%) | 161 (32.6%) | 463 (29.5%) |
| **Nodule size[b]** | | | | |
| Median | 38 | 35 | 37 | 35 |
| IQR (Q3-Q1) | 49.0–28.0 | 45.0–25.0 | 48.9–27.0 | 48.0–25.0 |
| <40 | 210 (53.3%) | 62 (62.0%) | 272 (55.1%) | 893 (57.0%) |
| ≥40 | 182 (46.2%) | 38 (38.0%) | 220 (44.5%) | 667 (42.5%) |

[a]Data files include biological replicates.
[b]Nodule size information of eight patients was not recorded.
*FFPE* formalin-fixed paraffin-embedded, *FNA* fine needle aspiration, *FTA* follicular thyroid adenoma, *FTC* follicular thyroid carcinoma, *IQR* interquartile range.

dimensionality reduction analysis, we observed that the two groups of samples could be partially distinguished by DEPs, but the overall discriminability remains low, which is closely reflects the histological similarities between the two (Fig. EV2C). Therefore, we further performed machine learning to improve discriminability.

After the performance comparison of six machine learning models with different feature counts (Appendix Fig. S2), the 24-protein-based XGBoost model was selected and applied in the following studies. Detailed model construction procedures and feature importance rank are shown in Fig. EV3. Our model achieved AUC values of 0.953 (95% CI, 0.936–0.971), 0.905 (95% CI, 0.886–0.915) and 0.899 (95% CI, 0.849–0.949) in the training, cross-validation and independent testing sets, respectively (Fig. 2B). In detail, the sensitivity, specificity, positive predictive values (PPV), negative predictive value (NPV), and accuracy were 0.800 (95% CI, 0.685–0.880), 0.843 (95% CI, 0.738–0.911), 0.825 (95% CI, 0.711–0.901), 0.819 (95% CI, 0.713–0.892), 0.822 (95% CI, 0.748–0.878), respectively, in the independent testing set (Table 2 and Fig. EV3C), which is much higher than the gene-based model. Next, we compared the model performance using gene-, TMT-protein- and combined feature pool. After 100 iterations of cross-validation testing, the model performance did not significantly improve with the addition of gene features on TMT-protein (Fig. 2C), indicating that the additive role of genetic features is minimal. The above results illustrate the significant potential for discriminating FTA and FTC within retrospective samples. Therefore, in subsequent studies, we further developed clinically accessible protein-based classification measurements.

## Targeted proteomics-based model development and evaluation

The previous in-depth proteomics approach is relatively costly and time-consuming for clinical diagnostic laboratories. In contrast, targeted proteomics, with its higher accuracy, stability, shorter run times, and cost-effectiveness, is more suitable for clinical application. Hence, we developed a targeted proteomic strategy for the protein biomarker candidates. A total of 44 proteins out of 187 DEPs were successfully detected using a single-injection targeted method. These proteins were further measured in four datasets (*n* = 1214) comprising retrospective FFPE and prospective FNA samples from 21 clinical centers (Fig. 1).

We initially analyzed 1054 samples from 18 centers, which were randomly allocated into discovery (*n* = 729) and testing (*n* = 325) sets. In the discovery set, the XGBoost algorithm was conducted and a panel of 24 protein biomarkers emerged and were ranked (Fig. EV4). The characteristics of each selected protein with corresponding abundance are shown in Appendix Table S2 and Appendix Fig. S3. Of the 24 selected proteins, eight (CA4, ITIH5, FABP4, DPP4, CRABP1, HMGA2, TIMP1, ECM1) are known to be associated with follicular thyroid tumors, ten (MATN2, AHSG, CD36, STMN1, NPC2, IGF2BP2, P4HA2, LRP2, IGSF1, RAP1-GAP) are related to other types of thyroid cancer, and the remaining six (CPOX, MYEF2, STEAP4, H1-5, FRAS1, TANC2) are newly discovered and have not been previously reported. After fine-tuning the hyperparameters, our model correctly identified 255 out of 325 samples with an accuracy of 0.785 (95% CI,

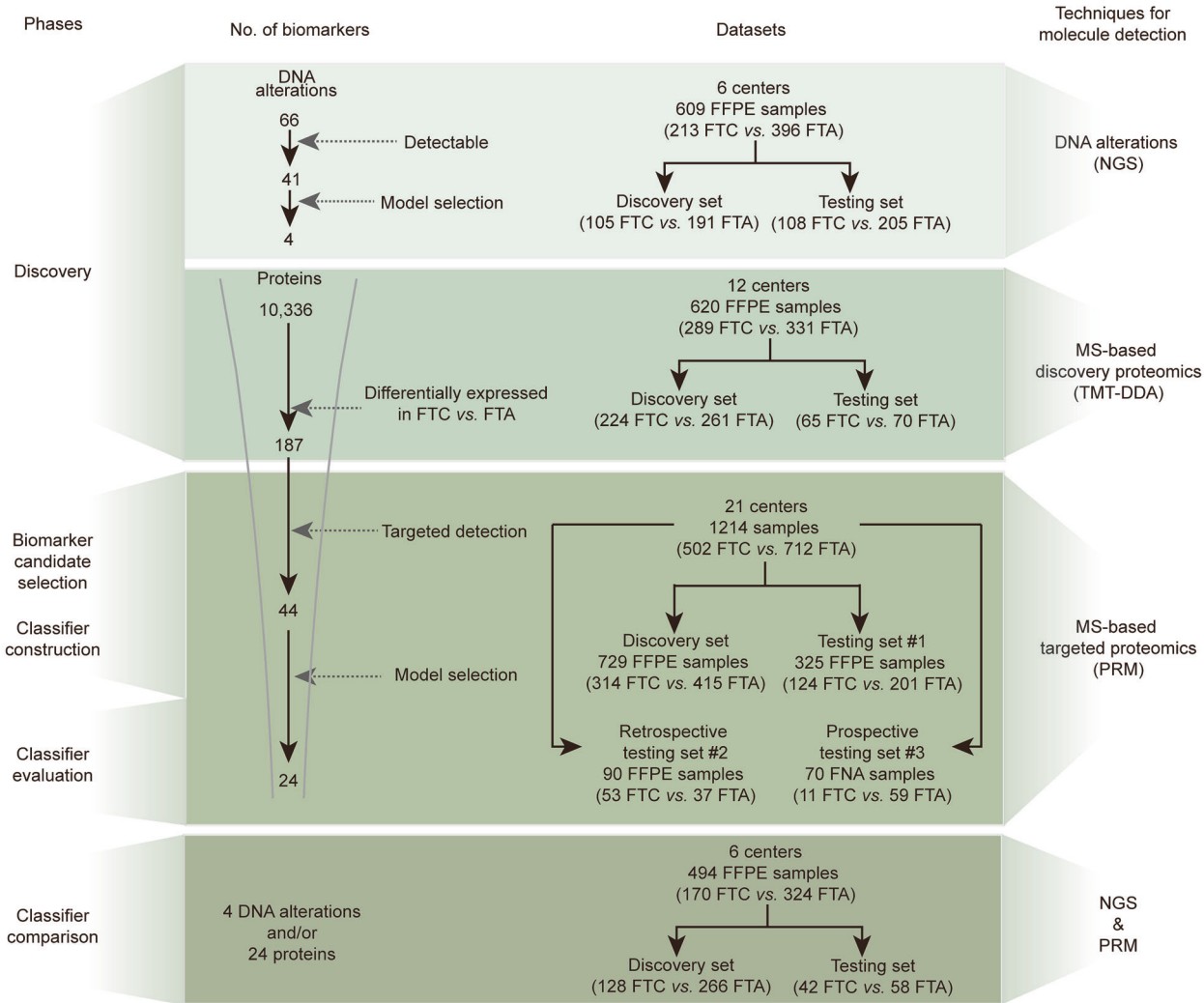

**Figure 1. The flowchart of biomarker discovery, classifier development, performance evaluation, and comparison.**

The entire experiment was conducted in the following phases: discovery of dysregulated molecules, biomarker candidate selection, classifier construction, classifier evaluation, and comparison. The molecules and samples involved at different stages are illustrated in the corresponding phases.

0.736–0.826), corresponding to a sensitivity of 0.726 (95% CI, 0.641–0.797) and a specificity of 0.821 (95% CI, 0.761–0.868). The PPV and NPV were 0.714 (95% CI, 0.630–0.786) and 0.829 (95% CI, 0.770–0.875), respectively, in the testing set (Table 2).

To further validate the generalization of our classifier, we additionally collected and tested two independent sample sets from international multicenters, one of which was retrospectively acquired, and the other prospectively. In the retrospective testing set, the classifier accurately identified 78.9% of the samples corresponding to the sensitivity, specificity, PPV and NPV of 0.792 (95% CI, 0.663–0.881), 0.784 (95% CI, 0.625–0.888), 0.840 (95% CI, 0.711–0.918), 0.725 (95% CI, 0.570–0.839), respectively. We further tested the model on prospective FNA biopsies, which were obtained pre-surgery. The model also performed well, achieving a diagnostic accuracy of 0.757 (95% CI, 0.644–0.843)

with sensitivity, specificity, PPV and NPV of 0.818 (95% CI, 0.510–0.957), 0.746 (95% CI, 0.621–0.840), 0.375 (95% CI, 0.212–0.574), 0.957 (95% CI, 0.845–0.995), respectively (Table 2).

Additionally, our model achieved AUC values of 0.871 (95% CI, 0.833–0.910), 0.853 (95% CI, 0.772–0.934) and 0.781 (95% CI, 0.563–1.000) in the internal testing set, the independent retrospective and prospective testing sets, respectively (Fig. 2D).

The foregoing multicenter testing results demonstrate that our protein-based model conducted by targeted proteomics can aid in the differential diagnosis of thyroid follicular tumors. It's worth mentioning that this protein-based classifier enhances the accuracy of pre-surgical diagnoses for follicular thyroid tumors, which currently defies a definitive solution in clinical practice.

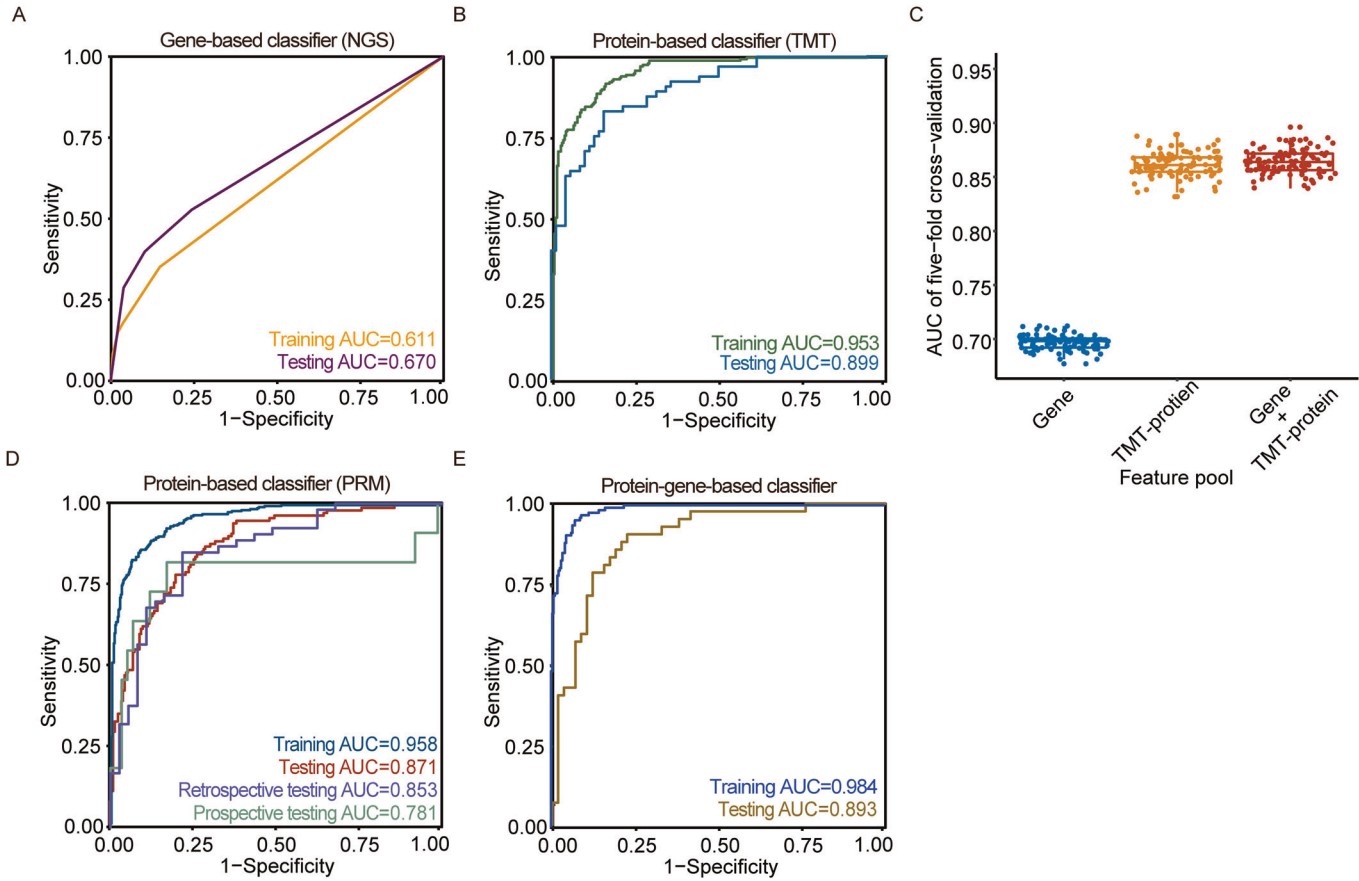

**Figure 2. Performance of models across different datasets.**

Area under the curve (AUC) plots for (A) four-gene-based classifier detected by next-generation sequencing (NGS), (B) 24-protein-based classifier detected by tandem mass tag (TMT) discovery proteomics, (D) 24-protein-based classifier detected by parallel reaction monitoring (PRM) targeted proteomics in retrospective and prospective sample sets, (E) four gene and 24 protein combination-based classifier. (C) The AUC comparison of gene-, TMT-protein-, gene and TMT-protein-based models of five-fold cross-validation for one-hundred-time iterations. Each dot indicates one iteration. The box plots (n = 100 per group) are defined as follows: the lower and upper edges of the box represent the 25th percentile (first quartile) and the 75th percentile (third quartile), respectively; the central line represents the 50th percentile (median); and the whiskers span from the 0th percentile (minimum) to the 100th percentile (maximum). Source data are available online for this figure.

## Comparative evaluation of four genes and/or 24 protein-based models

To equally compare the models based on genes, proteins, and the combination of both, we utilized the same set of samples from the overlapping samples in the two datasets (n = 494) with the same data splitting. The hyperparameters of each model were optimized to ensure the best performance. Based on 5-fold cross-validation, the AUC of combined gene and protein features was higher than only using 24 proteins (Fig. EV5A). Moreover, the higher AUC owed more to protein features than to gene features (Fig. EV5B). This model with combined features achieved an AUC of 0.893 (95% CI, 0.829–0.957) in the testing set (Fig. 2E). In detail (Table 2 and Fig. EV5C), the accuracy, sensitivity, specificity, positive predictive values (PPV) and negative predictive value (NPV) were 0.820 (95% CI, 0.732–0.883), 0.714 (95% CI, 0.563–0.829), 0.897 (95% CI, 0.788–0.954), 0.833 (95% CI, 0.676–0.924), 0.813 (95% CI, 0.698–0.890), respectively, in the independent testing set, which is comparable to the TMT-based model.

## Discussion

Although molecular tests have been adopted for the diagnosis of benign and malignant thyroid nodules (DiGennaro et al, 2022), differentiating FTC from FTA has long been a significant challenge in pathological diagnosis. This issue has persistently troubled clinicians and remains a particularly thorny problem. Despite numerous attempts over the years to distinguish between the two at various molecular levels, including DNA methylation (Yao et al, 2021; Zhang et al, 2021), mRNA (Makhlouf et al, 2016; Pfeifer et al, 2013; Wojtas et al, 2017), DNA-mRNA (Poma et al, 2018), and protein analyses (Castelblanco et al, 2017), no effective molecular markers or marker combinations have been identified to aid in clinical diagnosis.

Several factors may contribute to this limitation. First, there is insufficient detection depth. Since the boundary between FTA and FTC may not be well-defined and could represent different stages of the same disease, it is crucial to conduct deep, quantitative analyses to uncover subtle distinctions. Second, due to their inherent similarities, the identification of reliable molecular markers

depends heavily on large-scale data to enhance statistical reliability. Multicenter studies that incorporate both retrospective and prospective approaches can further improve the robustness of biomarker discovery.

Therefore, we designed an international multicenter study to explore the potential strategy for discriminating FTA and FTC. This international collaboration ensures a more comprehensive dataset, enhances the generalizability of the findings, and mitigates regional biases. Firstly, we assessed gene alterations in a large sample set containing 609 tissues the results of which were highly consistent with previous findings. Unlike high specificity (up to 98%) of *BRAF* V600E in papillary thyroid carcinoma (Kimura et al, 2003; Xing, 2005), no effective genetic markers have been identified to differentiate FTC and FTA, not even a panel of gene markers. Therefore, it is not surprising that our gene-based model could not distinguish the two. Proteins are the direct executors of biological activities, and some of them have been used as clinical biomarkers for various diseases, e.g., thyroglobulin (Tg) and thyroid peroxidase (TPO) for thyroid diseases, prostate-specific antigen (PSA) for prostate cancer, carcinoembryonic antigen (CEA) for liver cancer, and HER-2 for breast cancer subtyping. Based on these, we decided to compare the proteome of FTA and FTC through a deep proteome quantification strategy, namely TMT, for biomarker discovery and further developed a protein-based classifier by leveraging a clinically available targeted proteomic method, i.e., PRM.

We quantified over 10,000 proteins by TMT, as a high-quality dataset in the field of proteomics, laying a solid foundation for subsequent biomarker discovery. This quantification IDs are higher than those reported, 9826 (Sun et al, 2022b) and 7863 (Sun et al, 2022a) proteins quantified, in our previous studies. From the deep quantification matrix, we identified 187 potential DEPs. Next, 24 proteins filtered by XGBoost for high discriminant function achieved an AUC of 0.899. Compared to a gene-based model, a protein-based classifier has demonstrated significant potential for differentiation. Following this, we conducted targeted detection which can be adopted in the clinic. Targeted proteomics enables the rapid quantification of dozens to hundreds of proteins within a short time frame (typically from several minutes to tens of minutes), making it highly promising for clinical applications (Schiess et al, 2009). Similar methods are already in clinical use for detecting small molecules such as drugs, vitamins, and steroid hormones (Thomas et al, 2022). We next built a 24-protein-based model on a targeted proteomic matrix (44 proteins * 729 samples) and further validated it in two retrospective sets and a prospective testing set. The model achieved relatively high AUC of 0.871 and 0.853 in the retrospective sets. The lower AUC of 0.781 observed in FNA biopsies compared to FFPE samples can be attributed to the inherent sampling limitations and the heterogeneous nature of tumors. FNA biopsies, by their nature, sample only a small portion of the tumor, potentially missing areas of significant diagnostic or prognostic importance. Nevertheless, this protein-based classifier with a high NPV of 95.7% could improve diagnostic accuracy as a rule-out test to decrease unnecessary surgery on benign nodules.

The development of artificial intelligence offers more objective, accurate, and personalized options for medical evaluation (Gomes and Ashley, 2023; Yu et al, 2024), especially for complex diseases. Due to inter-individual heterogeneity, relying on a single or a few molecular markers can be susceptible to noise interference, making it difficult to obtain reliable results. In contrast, using a panel of

**Table 2. Performance evaluation of three classifiers on the corresponding testing sets.**

| Model | DNA-based | | | Protein-based (TMT) |
|---|---|---|---|---|
| Feature no. | 4 | | | 24 |
| Datasets | Testing set | | | Testing set |
| Ratio of malignancy | 0.345 | | | 0.4815 |
| AUC | 0.670 (0.612–0.729) | | | 0.899 (0.849–0.949) |
| Accuracy | 0.719 (0.666–0.766) | | | 0.822 (0.748–0.878) |
| Sensitivity | 0.407 (0.320–0.502) | | | 0.800 (0.685–0.880) |
| Specificity | 0.883 (0.831–0.920) | | | 0.843 (0.738–0.911) |
| PPV | 0.647 (0.528–0.750) | | | 0.825 (0.711–0.901) |
| NPV | 0.739 (0.680–0.790) | | | 0.819 (0.713–0.892) |
| **Model** | **Protein-based (PRM)** | | | **DNA&Protein-based** |
| Feature no. | 24 | | | 4 + 24 |
| Datasets | Testing set #1 | Testing set #2 | Testing set #3 | Testing set |
| Ratio of malignancy | 0.3815 | 0.5889 | 0.1571 | 0.42 |
| AUC | 0.871 (0.833–0.910) | 0.853 (0.772–0.934) | 0.781 (0.563–1.000) | 0.893 (0.829–0.957) |
| Accuracy | 0.785 (0.736–0.826) | 0.789 (0.692–0.861) | 0.757 (0.644–0.843) | 0.820 (0.732–0.883) |
| Sensitivity | 0.726 (0.641–0.797) | 0.792 (0.663–0.881) | 0.818 (0.510–0.957) | 0.714 (0.563–0.829) |
| Specificity | 0.821 (0.761–0.868) | 0.784 (0.625–0.888) | 0.746 (0.621–0.840) | 0.897 (0.788–0.954) |
| PPV | 0.714 (0.630–0.786) | 0.840 (0.711–0.918) | 0.375 (0.212–0.574) | 0.833 (0.676–0.924) |
| NPV | 0.829 (0.770–0.875) | 0.725 (0.570–0.839) | 0.957 (0.845–0.995) | 0.813 (0.698–0.890) |

Values (95% confidence interval).
*AUC* area under the curve, *PPV* positive predictive value, *NPV* negative predictive value.

proteins (typically dozens) effectively mitigates this issue (Mann et al, 2021). In this study, we balanced performance and the number of protein features, optimizing between 20–30 proteins, and ultimately identified 24 proteins to incorporate into the model. While proteins can be detected using antibody-based methods, clinical immunohistochemistry (IHC) typically allows for the detection of only one protein per assay, with the number of detectable proteins limited by the number of tissue sections (Guo et al, 2025). Although multi-color IHC is extensively used in scientific research, it has not yet been broadly adopted in real-world clinical diagnostics. Moreover, antibody-based quantitative detection is constrained by the specificity, sensitivity, and linear range of the antibodies, and is prone to be affected by experimental conditions and reagent batches. MS, on the other hand, offers high-throughput, high-precision, diverse, and consistent protein quantification (Aebersold et al, 2013; Wiśniewski and Mann, 2016), making it more compatible with machine learning techniques to establish complex and reliable models for disease classification, diagnosis, and other applications. It is anticipated that MS will play an indispensable role in future clinical diagnostics and treatment.

Although we have achieved promising results, there are still some limitations that need to be addressed. First, there could be a sampling bias in this retrospective study, which may have affected the model performance in prospective testing set. Second, in the current model construction and evaluation, it is challenging to completely rule out false-negative samples, particularly those in the early stages of FTC without capsular invasion at the time of surgery. Third, the small number of patients of racial or ethnic minority groups included in this study may limit generalizability to these underrepresented groups.

In summary, this discovery investigation with subsequent prospective validation, shows that integrating deep proteomics and targeted proteomics coupled with machine learning facilitates precise diagnosis of follicular thyroid tumors. This paradigm can also be extended to the differential diagnosis of other types of diseases in the future.

# Methods

### Reagents and tools table

| Reagent/Resource | Reference or Source | Identifier or Catalog Number |
|---|---|---|
| **Chemicals, enzymes and other reagents** | | |
| Triethylammonium bicarbonate (TEAB) | Sigma Aldrich | T7408 |
| Heptane | Sigma Aldrich | 246654 |
| Tris base | Sigma Aldrich | 252859 |
| Urea | Sigma Aldrich | U1250 |
| Thiourea | Sigma Aldrich | T8656 |
| Iodoacetamide (IAA) | Sigma Aldrich | I6125 |
| Tris (2-carboxyethyl) phosphine (TCEP) | Sigma Aldrich | 61820E |
| Trypsin | Hualishi Tech | HLS rTRY001C |
| Lys-C | Hualishi Tech | HLS rLYSC |
| TMTpro 16plex reagents | Thermo Fisher Scientific | A44520 |

| Reagent/Resource | Reference or Source | Identifier or Catalog Number |
|---|---|---|
| Hydroxylamine | Thermo Fisher Scientific | 90115 |
| LC-MS-grade water | Thermo Fisher Scientific | W6-4 |
| Trifluoroacetic acid (TFA) | Thermo Fisher Scientific | 85183 |
| Formic acid (FA) | Thermo Fisher Scientific | A117-50 |
| Acetonitrile | Thermo Fisher Scientific | A955-4 |
| Methanol | Sigma Aldrich | 34860 |
| Ammonium hydroxide solution | Sigma Aldrich | 221228 |
| QIAamp DNA FFPE Advanced Kit | QIAGEN | 56604 |
| Unique dual index and Illumina sequencing adapters | Shanghai Rigen Biotechnology | RJ013J-A, B, C, D, E, F |
| Thyroid cancer-related 66-gene panel | Shanghai Rigen Biotechnology | RJ015J |
| Qubit dsDNA HS Assay Kit | Thermo Fisher Scientific | Q32854 |
| NovaSeq 6000 S4 v1.5 Kit | Illumina | 20028312 |
| **Software** | | |
| R (version 4.2.3) | The R project | https://posit.co/download/rstudio-desktop/ |
| Protein Discoverer (version 2.4) | Thermo Fisher Scientific | https://www.thermofisher.com/order/catalog/product/OPTON-31014 |
| Skyline (version 23.1) | (MacLean et al, 2010) | https://skyline.ms/project/home/software/Skyline/begin.view |
| **Other** | | |
| Qubit fluorometer | Thermo Fisher Scientific | Q33216 |
| NovaSeq 6000 | Illumina | 20012850A |
| Pressure cycling technology | PBI | N/A |
| Thermo Ultimate Dionex 3000 | Thermo Fisher Scientific | N/A |
| XBridge Peptide BEH C18 column | Waters | N/A |
| Orbitrap Exploris 480 | Thermo Fisher Scientific | N/A |
| FAIMS Pro Duo interface | Thermo Fisher Scientific | N/A |

## Patients and samples

This study protocol and waiver of informed consent were approved by the Ethics Committee of Westlake University with approval No. 20240708GTN001 and all study methodologies adhered to the guidelines

outlined in the Declaration of Helsinki. Our study sets were collected from 24 participating centers of the Westlake Thyroid Proteome Consortium (WE-TEC) working group. Clinical demographic data and histopathological reports were extracted from each medical record system. Histopathology was reviewed for hematoxylin and eosin (H&E) slides according to the standardized WHO classification (4th and 5th editions) (Baloch et al, 2022; Lloyd et al, 2017). Based on the pathological results, we included FTA and FTC samples and excluded samples with follicular variant papillary thyroid tumors.

In the retrospective sets, formalin-fixed paraffin-embedded (FFPE) slides or punches from 2002 to 2022 were collected. Additionally, we included 70 fine needle aspiration (FNA) biopsies with histopathology reports of follicular thyroid neoplasm from our registered prospective cohort from eight centers.

The sample datasets comprised four main parts. Firstly, there were 609 retrospective FFPE samples designated for DNA mutation detection. Secondly, 645 retrospective FFPE samples from 12 centers were used for protein biomarker discovery. Thirdly, 1054 retrospective FFPE samples were analyzed for targeted proteome analysis and classifier development. Lastly, the dataset included 90 retrospective FFPE samples and 70 prospective fine needle aspiration biopsies, which were utilized for independent model testing.

The sample size was determined by the available specimens at study initiation without prior sample size calculations. To minimize artificial effects, all samples within each dataset were randomly allocated to processing batches. Sample labels remained blinded throughout the entire workflow—including sample processing, mass spectrometry data acquisition, and model testing—ensuring unbiased evaluation.

## Gene mutation analysis

Genomic DNA was extracted from thyroid FFPE slides utilizing the QIAamp DNA FFPE Advanced Kit (Cat. ID 56604, QIAGEN, Germany). Sequencing libraries were constructed using a designed thyroid cancer-related 66-gene panel (Appendix Table S3, Shanghai Rigen Biotechnology, China), which is designed for multiplex targeted sequencing, enabling the detection of point mutations and insertions/deletions. An aliquot of extracted DNA was used for multiplex amplification of target regions, followed by polymerase chain reaction (PCR) amplification to add unique dual index and Illumina sequencing adapters (Shanghai Rigen Biotechnology, China). Post amplification, the indexed libraries were purified with beads, quantified using a Qubit fluorometer (Thermo Fisher Scientific, USA), and sequenced to generate 150 bp paired-end reads on the NovaSeq 6000 platform (Illumina Inc., USA).

Quality assessment of the raw sequencing data was performed using FastQC (version 0.11.9). Adapter sequences and low-quality bases were trimmed from the raw reads using Cutadapt (version 1.18) (Martin, 2011). The trimmed reads were then aligned to the human reference genome (hg19) with BWA (version 0.7.17) (Li and Durbin, 2009). Variants, including single-nucleotide variants (SNVs) and insertions/deletions (InDels), were identified using VarScan2 (version 2.4.4) (Koboldt et al, 2012), and subsequently annotated using the Ensembl Variant Effect Predictor (McLaren et al, 2016).

## Sample processing for proteome analysis

FFPE tissue slides and FNA biopsies were processed using the pressure cycling technology (PCT)-assisted sample preparation pipeline as described in our previous publications (Cai et al, 2022; Gao et al, 2020). Briefly, the FFPE samples were dewaxed with heptane and rehydrated using ethanol solutions of various concentrations. Following this, the samples underwent acidic hydrolysis with 0.1% formic acid and basic hydrolysis with 0.1 M Tris-HCl (pH 10.0). Proteins were then extracted using a lysis buffer containing 6 M urea and 2 M thiourea. The proteins were reduced and alkylated with 10 mM Tris (2-carboxyethyl) phosphine and 40 mM iodoacetamide under PCT assistance. Subsequently, PCT-assisted Lys-C/trypsin enzymatic digestion was performed, with the optimal enzyme-to-substrate ratios of 1:80 for Lys-C and 1:20 for trypsin.

TMT-based proteomics analysis was performed as previously described (Nie et al, 2021). Briefly, 7 μg of peptides from each tissue sample and pooled peptide sample were labeled with the TMTpro 16-plex reagent (Thermo Fisher Scientific, San Jose, USA). Hydroxylamine was employed to quench the labeling reaction. Subsequently, the 16 TMT-labeled peptide samples were combined and cleaned using C18 columns. High-pH fractionation was then performed on an UltiMate™ Dionex 3000 (Thermo Fisher Scientific, San Jose, USA) equipped with an XBridge Peptide BEH C18 column (300 Å, 5 μm, 4.6 mm × 250 mm) (Waters, Milford, MA, USA). The fractionation process utilized a 120-min LC gradient, ranging from 5% to 35% acetonitrile (ACN) in 10 mM ammonia (pH 10.0), at a flow rate of 1 mL/min, resulting in 120 fractions. As previously described (Nie et al, 2021), these 120 fractions were then combined into 30 fractions.

## Discovery proteomics data acquisition and processing

Peptides from each fraction were analyzed by an Orbitrap Exploris™ 480 mass spectrometry (Thermo Fisher Scientific, San Jose, USA) coupled with FAIMS Pro Duo interface (Thermo Fisher Scientific, San Jose, USA), along with a 60 min LC gradient at a flow rate of 300 nL/min. Subsequently, the fractionated sample was separated with Thermo Scientific UltiMate™ 3000 RSLCnano System. The mass spectrometer was operated in positive mode with the FAIMS Pro interface and then analyzed with the mass spectrometry by data-dependent acquisition (DDA) mode. The compensation voltage was set to −45 V and −65 V with a cycle time of 1 s per FAIMS experiment.

All the DDA data were processed using Proteome Discoverer version 2.4 (Thermo Scientific, USA) against a FASTA file downloaded from UniProt database (version 15/07/2020, 20368). The settings were referred to in a previously published paper. Briefly, missed cleavages within two were allowed. The minimal peptide length was set as six residues. Normalization was processed against the total peptide amount. Precursor ion mass tolerance was set to 10 ppm, and fragment mass tolerance was 0.02 Da. The false discovery rate (FDR) of peptides was set to 1% (strict) and 5% (relaxed).

## Data acquisition and processing of parallel-reaction monitoring (PRM)

PRM detection was performed preliminarily based on 187 selected proteins which were chosen from differentially expressed proteins (DEPs) in the TMT dataset and our previous published datasets (Sun et al, 2022a; Sun et al, 2022b). After optimization, a set of 44 proteins from the original panel of 187 proteins. For each protein,

we select one precursor to be monitored by Skyline (MacLean et al, 2010) (version 23.1) from our established thyroid-specific spectral library (Li et al, 2024a). The selection criteria are as follows: (a) no peptide modification, (b) no missed cleavages, and (c) peptide length ranging from 8 to 20.

Cleaned peptides were separated through UltiMate™ 3000 RSLCnano System (Thermo Fisher Scientific, San Jose, CA) equipped with a 15 cm × 75 μm analytical column (1.9 μm 100 Å C18-Aqua) through a 60-min effective linear gradient of 6% to 30% buffer B (98% ACN, 0.1% formic acid) at 300 nL/min. The separated peptides were further analyzed by Q Exactive™ HF (Thermo Fisher Scientific, San Jose, CA) with PRM data acquisition mode.

The time-scheduled acquisition mode was applied within a $+/-3$ min retention time window. The MS1 scans were collected $m/z$ at 400 to 2500 Th with a resolution of 60,000 FWHM. The AGC target was set to 3E6 charges and the maximum IT was 55 ms. The target precursors were isolated through a window $m/z$ of 1.6 Th. The normalized collision energy for fragmentation was set at 27%. The products were scanned at a resolution of 30,000 FWHM, the AGC target value was set to 5E5 charges, and the maximum injection time was 120 ms. Here, a total of 44 precursors from 44 proteins and 20 CiRT were analyzed. PRM data were further analyzed by Skyline (version 23.1) with the same setting when we developed the PRM method. Next, the protein abundance matrix was transformed by log2.

## TMT data quality control and preprocessing

We firstly excluded ten samples, of which two samples are not follicular tumors, six samples are from metastatic sites, and two samples have incomplete clinical information. For the pooled samples, data quality was assessed by analysis of the coefficients of variation (CV) across pooled samples. Next, principal component analysis (PCA) was performed based on the discovery set with missing values imputed with zero. We next deleted fifteen high-missing-proportion samples which were detected as outliers in the PCA. We ultimately analyzed a total of 620 FFPE tissue slides comprising 331 FA and 289 FTC from 12 clinical centers. Additionally, the proteins with NA rate >60% were removed.

Missing values imputation was conducted by R package *NAguideR* (Wang et al, 2020) and the impseqrob algorithm was used. To avoid leakage of testing data, we imputed the discovery set first, which ensured the imputation for testing data did not affect the discovery set. Then, the testing set was combined with the filled discovery set and imputed by the same imputation method impseqrob.

Batch effects were detected by two-dimensional uniform manifold approximation and projection (UMAP) visualization. Batch effects were corrected by the Combat algorithm in R package *sva* (Leek et al, 2012), which is an empirical Bayes framework for adjusting data for batch effects. Similarly, we adjusted the batch effects of the discovery set first. As for the testing set, to prevent testing data or label leakage, we held the testing set label unknown for Combat and corrected batch effects referring to the discovery set after correction, keeping the discovery set unchanged.

## PRM data preprocessing

For PRM data, the Deterministic Minimal Value (MinDet) algorithm was first utilized for imputing the missing values in the

matrix. For the retrospective and prospective testing sets, the ComBat algorithm was selected to deal with the batch effects, with the PRM training set as the reference dataset.

## Modeling

### Gene-mutation-based modeling

The establishment of a gene-mutation-based classifier was based on the dataset of 609 patients from six centers. Patients from one center were selected as training sets and applied for model building. Samples from the other five centers were adopted for independent testing. Gene data in the analyzed matrix were transformed to logical data where 0 and 1 represent wide type and mutation, respectively.

### TMT-based protein modeling

To build and evaluate our classification system, we split our data into two parts: a discovery set ($n = 485$) and an independent testing set ($n = 135$).

The machine learning-related contents were implemented by R package *mlr3*, which is a modern object-oriented machine learning framework in R. Before building machine learning models, to simplify the feature selection and engineering, we chose to narrow down the feature set by ensemble filtering. In detail, three filtering criteria, ANOVA $P$ value, Kruskal-Wallis test $P$ value and information gain were considered and features with ANOVA $P$ values no less than 0.001 or Kruskal-Wallis test $P$ values no less than 0.001 or no information gain would be removed.

Six machine learning algorithms (Lang et al, 2019), K-Nearest Neighbor (KNN), Naïve Bayes (NB), Logistic regression (L), Support Vector Machine with radial basis function kernel (SVM), Random Forest (RF), and eXtreme Gradient boosting (XG or XGboost) were chosen as potential algorithms to deal with the FA and FTC classification task. Before benchmarking these algorithms, we tuned the hyper-parameters of the 6 machine learning models by random search (100 hyper-parameters combinations tried) and 5-fold cross-validation on the discovery set and determined the best hyper-parameter setting. Then to compare the models under the best hyper-parameter setting, we ranked the features by averaging the rankings of three filter criteria, used different feature number settings and conducted 100 times 5-fold cross-validation for each model under each feature number setting with AUC recorded. The best model was determined by referring to the average AUCs under different feature numbers and would be used for downstream analysis.

To refine the selected model, a more concise and model-based feature selection was conducted. In detail, we first trained XGboost on a discovery set 100 times with different random seeds, each time selecting the 50 most important features and retaining the features that appeared no less than 30 times. Additionally, the feature importance mentioned above was calculated by gain which represents the fractional contribution of each feature to the model based on the total gain of this feature's splits and a higher percentage means a more important predictive feature.

### PRM-based protein modeling

For PRM modeling, the proteins with fold change >1.4 and Benjamini & Hochberg (BH)-adjusted Welch's $t$-test $P$ value < 0.05 were selected as important features for further modeling. For XGBoost model building, the hyperparameters (lambda, alpha, and nrounds) were

**The paper explained**

**Problem**

Distinguishing follicular thyroid adenoma (FTA) from carcinoma (FTC) remains challenging. Current diagnosis requires postoperative histopathology to identify capsular invasion, which cannot be assessed preoperatively. Even experienced pathologists struggle with this differentiation, and existing molecular markers cannot reliably distinguish between these conditions.

**Results**

Using 2443 thyroid samples from 1568 patients (909 FTA, 659 FTC) across 24 centers in China and Singapore, we performed next-generation sequencing with a 66-gene panel and mass spectrometry-based proteomics. While 41 of 66 target genes (62.1%) were detected with similar alteration patterns between FTA and FTC, discovery proteomic analysis identified 10,336 proteins with 187 dysregulated. XGBoost machine learning models revealed the protein-based approach significantly outperformed the gene-based model (AUROC 0.899 [95% CI, 0.849–0.949] vs. 0.670 [95% CI, 0.612–0.729]). Our 24-protein classifier maintained high performance across retrospective cohorts (AUC 0.871 [95% CI, 0.833–0.910] and 0.853 [95% CI, 0.772–0.934]) and prospective fine-needle aspiration biopsies (AUC 0.781 [95% CI, 0.563–1.000]), with an impressive 95.7% negative predictive value for ruling out malignancy.

**Impact**

This study illustrates that integrating deep proteomics and targeted proteomics with machine learning enables precise diagnosis of follicular thyroid tumors, reducing unnecessary surgeries. The established paradigm has potential for broader application in differential diagnosis across various diseases, representing a significant advancement in precision medicine.

the BH method. The confidence intervals for training and testing performance measures except for AUC were Wilson's 95% confidence interval, and the variance of the AUC was computed as defined by the bootstrap method by the pROC package and the 95% confidence interval was deduced with the normal distribution. As for the confidence intervals of cross-validation performance measures, they were approximately 95% confidence intervals from mean $- 1.96 *$standard deviation to mean $+ 1.96 *$standard deviation. The cross-validation ROC originated from averaging five ROC curves in 5-fold cross-validation.

## Data availability

All the proteomic raw data have been deposited to the ProteomeXchange Consortium (http://proteomecentral.proteomexchange.org) under the identifier IPX0008384000. Specifically, the TMT, PRM discovery, and PRM testing raw files are available under the identifiers IPX0008384001, IPX0008384003, and IPX0008384002, respectively. The code for statistical analysis and modeling presented in this manuscript and generating corresponding figure panels and tables are publicly available on GitHub at https://github.com/guomics-lab/MFT.

The source data of this paper are collected in the following database record: biostudies:S-SCDT-10_1038-S44321-025-00242-2.

## Peer review information

tuned by 5-fold cross-validation and random search based on the training set, also towards the highest classification AUC.

### Modeling based on gene and PRM-based protein data

For Gene and PRM modeling, firstly, we only retained the four feature genes and 24 feature proteins that were selected by previous models. The applied dataset ($n = 494$) contains both gene-mutation and PRM-based protein data. The training set ($n = 394$) for this comparison was derived from the previously established model training set combination, and the remaining samples were designated as the testing set ($n = 100$). Next, we conducted 100 times 5-fold cross-validation to compare gene-based, PRM-based and gene & PRM-based modeling. For XGBoost model building, the hyperparameters (lambda, alpha, and nrounds) were tuned by 5-fold cross-validation and random search based on the training set, also towards the highest classification AUC.

## Statistical analysis

The statistical and bioinformatic analyses were conducted by R (version 4.2.3). PCA in the quality control part was performed based on a centered but not scaled discovery set with missing values imputed by zero. UMAP was implemented by function umap in R package *umap* with default parameter setting. The Welch's *t*-test was utilized to compare the expression difference of proteins between FA and FTC and the resulting *p*-values were adjusted by

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

## Acknowledgements

This work is supported by the Noncommunicable Chronic Diseases-National Science and Technology Major Project (No. 2024ZD0525603 to Haixia Guan), National Key R&D Program of China (No. 2022YFF0608403 to Yi Zhu) and (No. 2021YFA1301600 to Tiannan Guo), "Pioneer" and "Leading Goose" R&D Program of Zhejiang (No. 2024SSYS0035 to Tiannan Guo), China Postdoctoral Science Foundation (No. 2022M722841 to Yaoting Sun), and the State Key Laboratory of Medical Proteomics. We thank the Westlake University Supercomputer Centre for data storage and computation. Gene alteration detection was supported by RIGEN Biotechnology Co., Ltd, special appreciation goes to Dr. Wentian He and Dr. Yuxin Li for their valuable insights on gene analysis.

## Author contributions

**Yaoting Sun**: Conceptualization; Data curation; Formal analysis; Funding acquisition; Investigation; Visualization; Methodology; Writing—original draft; Project administration; Writing—review and editing. **He Wang**: Formal analysis; Methodology; Writing—original draft; Writing—review and editing. **Lu Li**: Data curation; Investigation; Visualization; Writing—review and editing. **Jianbiao Wang**: Resources; Investigation. **Wanyuan Chen**: Resources; Investigation. **Li Peng**: Resources; Investigation. **Pingping Hu**: Data curation; Investigation. **Jing Yu**: Data curation; Investigation. **Xue Cai**: Data curation; Investigation. **Nan Yao**: Data curation; Investigation. **Yan Zhou**: Data curation; Investigation. **Jiatong Wang**: Data curation; Investigation. **Yingrui Wang**: Writing—review and editing. **Liqin Qian**: Writing—review and editing. **Weigang Ge**: Investigation. **Mengni Chen**: Investigation. **Feng Yang**: Data curation. **Zhiqiang Gui**: Data curation. **Wei Sun**: Resources. **Zhihong Wang**: Resources. **Minghua Ge**: Resources. **Yi He**: Resources. **Guangzhi Wang**: Resources. **Yongfu Zhao**: Resources. **Huanjie Chen**: Resources. **Xiaohong Wu**: Resources. **Yuxin Du**: Resources. **Wenjun Wei**: Resources. **Fan Wu**: Resources. **Dingcun Luo**: Resources. **Xiangfeng Lin**: Resources. **Haitao Zheng**: Resources. **Xin Zhu**: Resources. **Bei Wei**: Resources. **Jiafei Shen**: Resources. **Jincao Yao**: Resources.

**Zhennan Yuan**: Resources. **Tong Liu**: Resources. **Jun Pan**: Resources. **Yifeng Zhang**: Resources. **Yangfan Lv**: Resources. **Qiaonan Guo**: Resources. **Qijun Wu**: Resources. **Tingting Gong**: Resources. **Ting Chen**: Resources. **Shu Zheng**: Resources. **Jingqiang Zhu**: Resources. **Hanqing Liu**: Resources. **Chuang Chen**: Resources. **Hong Han**: Resources. **Sathiyamoorthy Selvarajan**: Resources. **Michael Mingzhao Xing**: Writing—review and editing. **Kennichi Kakudo**: Writing—review and editing. **Erik K Alexander**: Writing—review and editing. **Yijun Wu**: Resources. **Yu Wang**: Resources. **Dong Xu**: Resources. **Hao Zhang**: Resources. **Xiu Nie**: Resources. **Oi Lian Kon**: Writing—review and editing. **N Gopalakrishna Iyer**: Resources. **Zhiyan Liu**: Resources. **Yi Zhu**: Conceptualization; Resources; Supervision; Funding acquisition; Writing—review and editing. **Haixia Guan**: Conceptualization; Resources; Supervision; Funding acquisition; Writing—review and editing. **Tiannan Guo**: Conceptualization; Resources; Supervision; Funding acquisition; Writing—review and editing. **We-TEC Investigators**: Support the project as a consortium.

Source data underlying figure panels in this paper may have individual authorship assigned. Where available, figure panel/source data authorship is listed in the following database record: biostudies:S-SCDT-10_1038-S44321-025-00242-2.

## Disclosure and competing interests statement

T Guo and Y Zhu are shareholders of Westlake Omics Inc. MC and WG are employees of Westlake Omics Inc. FY is a shareholder of the RIGEN Biotechnology Co., Ltd. T Guo, YS, and HW have applied for a patent [ID: ZL 2022 1 1046085.8] on this project. The other authors declare no competing interests in this paper.

[1]Affiliated Hangzhou First People's Hospital, State Key Laboratory of Medical Proteomics, School of Medicine, Westlake University, No. 18 Shilongshan Road, Hangzhou 310024, China. [2]Westlake Centre for Intelligent Proteomics, Westlake Laboratory of Life Sciences and Biomedicine, No. 600 Dunyu Road, Hangzhou 310030, China. [3]Research Centre for Industries of the Future, School of Life Sciences, Westlake University, No. 600 Dunyu Road, Hangzhou 310030, China. [4]College of Pharmaceutical Sciences, Zhejiang University, No. 866 Yuhangtang Road, Hangzhou 310058, China. [5]Department of Head and Neck Surgery, The Affiliated Sir Run Run Shaw Hospital, School of Medicine, Zhejiang University, No. 3 East Qingchun Road, Hangzhou 310016, China. [6]Department of Pathology, Zhejiang Provincial People's Hospital (Affiliated People's Hospital, Hangzhou Medical College), No. 158 Shangtang Road, Hangzhou 310006, China. [7]Department of Pathology, Union Hospital, Tongji Medical College, Huazhong University of Science and Technology, No. 1277 Jiefang Road, Wuhan 430022, China. [8]Westlake Omics (Hangzhou) Biotechnology Co. Ltd, No. 1 Yunmeng Road, Hangzhou 310024, China. [9]RIGEN Biotechnology Co. Ltd., No. 3632 Zhaolou Road, Shanghai 201102, China. [10]Department of Thyroid Surgery, The First Hospital of China Medical University, No. 155 Nanjingbei Road, Shenyang 110001, China. [11]Otolaryngology & Head and Neck Centre, Cancer Centre, Department of Head and Neck Surgery, Zhejiang Provincial People's Hospital (Affiliated People's Hospital, Hangzhou Medical College), No. 158 Shangtang Road, Hangzhou 310006, China. [12]Department of Urology, The Second Hospital of Dalian Medical University, No. 467 Zhongshan Road, Dalian 116027, China. [13]Department of General Surgery, The Second Hospital of Dalian Medical University, No. 467 Zhongshan Road, Dalian 116027, China. [14]Department of Thyroid Surgery, Qingdao Municipal Hospital, No. 1 Jiaozhou Road, Qingdao 266011, China.

[15]Department of Endocrinology, Zhejiang Provincial People's Hospital (Affiliated People's Hospital, Hangzhou Medical College), No. 158 Shangtang Road, Hangzhou 310006, China. [16]Department of Head and Neck Surgery, Fudan University Shanghai Cancer Centre; Department of Oncology, Shanghai Medical College, Fudan University, No. 270 Dongan Road, Shanghai 200032, China. [17]Department of Surgical Oncology, Affiliated Hangzhou First People's Hospital, School of Medicine, Westlake University, Hangzhou 310006, China. [18]Department of Thyroid Surgery, The Affiliated Yantai Yuhuangding Hospital of Qingdao University, No. 20 East Yuhuangding Road, Yantai 264000 Shandong, China. [19]Key Laboratory of Head and Neck Cancer Translation Research of Zhejiang Province, Zhejiang Cancer Hospital, No. 38 Guangji Road, Hangzhou 310022, China. [20]Zhejiang Cancer Hospital, Hangzhou Institute of Medicine (HIM), Chinese Academy of Sciences, No. 38 Guangji Road, Hangzhou 310022, China. [21]Department of Oncology Surgery, Harbin Medical University Cancer Hospital, No. 150 Haping Road, Harbin 150081, China. [22]Harbin Medical University Cancer Hospital, No. 150 Haping Road, Harbin 150081, China. [23]Department of Thyroid Surgery, The First Affiliated Hospital, School of Medicine, Zhejiang University, No. 79 Qingchun Road, Hangzhou 310003, China. [24]Department of Medical Ultrasound, Shanghai Tenth People's Hospital, Tongji University School of Medicine, No. 36 Yunxin Road, Shanghai 200072, China. [25]Department of Pathology, Xinqiao Hospital, Third Military Medical University (Army Medical University), No. 83 Xinqiao Road, Chongqing 400037, China. [26]Department of Clinical Epidemiology, Shengjing Hospital of China Medical University, No. 36 Sanhao Road, Shenyang 110004, China. [27]Department of Obstetrics and Gynecology, Shengjing Hospital of China Medical University, No. 36 Sanhao Road, Shenyang 110004, China. [28]Cancer Institute (Key Laboratory of Cancer Prevention and Intervention, China National Ministry of Education), The Second Affiliated Hospital, School of Medicine, Zhejiang University, No. 68 Jiefang Road, Hangzhou 310009, China. [29]Division of Thyroid Surgery, West China Hospital, Sichuan University, No. 37 Guoxuexiang, Chengdu 610041, China. [30]Department of Breast and Thyroid Surgery, Renmin Hospital of Wuhan University, No. 99 Zhangzhidong Road, Wuhan 430060, China. [31]Department of Ultrasound, Zhongshan Hospital, Institute of Ultrasound in Medicine and Engineering, Fudan University, No.180 Fenglin Road, Shanghai 200032, China. [32]Department of Anatomical Pathology, Division of Pathology, Singapore General Hospital, Qutram Road, Singapore 169608, Singapore. [33]School of Medicine, Southern University of Science and Technology, 1088 Xueyuan Avenue, Shenzhen, Guangdong 518055, China. [34]Department of Pathology, Cancer Genome Centre and Thyroid Disease Centre, Izumi City General Hospital, Izumi, Japan. [35]Thyroid Section, Brigham and Women's Hospital, Harvard Medical School, Boston, MA, USA. [36]Division of Medical Sciences, National Cancer Centre Singapore, 30 Hospital Boulevard, Singapore 168583, Singapore. [37]Department of Head and Neck Surgery, National Cancer Centre Singapore, 30 Hospital Boulevard, Singapore 168583, Singapore. [38]Department of Pathology, Shanghai Sixth People's Hospital Affiliated to Shanghai Jiao Tong University School of Medicine, No. 600 Yishan Road, Shanghai 200235, China. [39]Department of Endocrinology, Guangdong Provincial People's Hospital (Guangdong Academy of Medical Sciences), Southern Medical University, 106 Zhongshan erlu, Guangzhou 510080, China. [40]Affiliated Hangzhou First People's Hospital, School of Medicine, Westlake University, Hangzhou 310006, China. [41]These authors contributed equally: Yaoting Sun, He Wang, Lu Li, Jianbiao Wang, Wanyuan Chen, Li Peng. ✉E-mail: zhuyi@westlake.edu.cn; guanhaixia@gdph.org.cn; guotiannan@westlake.edu.cn

## WE-TEC INVESTIGATORS

Yaoting Sun (iD) [1,2,3,41], He Wang (iD) [1,2,3,41], Lu Li [1,2,3,4,41], Jianbiao Wang [5,41], Wanyuan Chen [6,41], Li Peng (iD) [7,41], Pingping Hu [1,2,3], Jing Yu [1,2,3], Xue Cai [1,2,3], Nan Yao [1,2,3], Yan Zhou [1,2,3], Jiatong Wang [1,2,3], Yingrui Wang (iD) [1,2,3], Liqin Qian [1,2,3], Weigang Ge (iD) [8], Mengni Chen (iD) [8], Feng Yang [9], Zhiqiang Gui [10], Wei Sun [10], Zhihong Wang [10], Minghua Ge [11], Yi He [12], Guangzhi Wang [13], Yongfu Zhao [13], Huanjie Chen [14], Xiaohong Wu [15], Yuxin Du [16], Wenjun Wei [16], Fan Wu [17], Dingcun Luo [17], Xiangfeng Lin [18], Haitao Zheng [18], Xin Zhu [19], Bei Wei [19,20], Jiafei Shen [19,20], Jincao Yao [19,20], Zhennan Yuan [21], Tong Liu [22], Jun Pan [23], Yifeng Zhang [24], Yangfan Lv [25], Qiaonan Guo [25], Qijun Wu [26], Tingting Gong [27], Ting Chen [28], Shu Zheng [28], Jingqiang Zhu [29], Hanqing Liu [30], Chuang Chen (iD) [30], Hong Han [31], Sathiyamoorthy Selvarajan [32], Michael Mingzhao Xing [33], Kennichi Kakudo (iD) [34], Erik K Alexander [35], Yijun Wu [23], Yu Wang [16], Dong Xu [19,20], Hao Zhang [10], Xiu Nie [7], Oi Lian Kon [36], N Gopalakrishna Iyer [37], Zhiyan Liu (iD) [38], Yi Zhu (iD) [1,2,3✉], Haixia Guan (iD) [39✉] & Tiannan Guo (iD) [1,2,3,40✉]

A full list of members and their affiliations appears in the Supplementary Information.

# Expanded View Figures

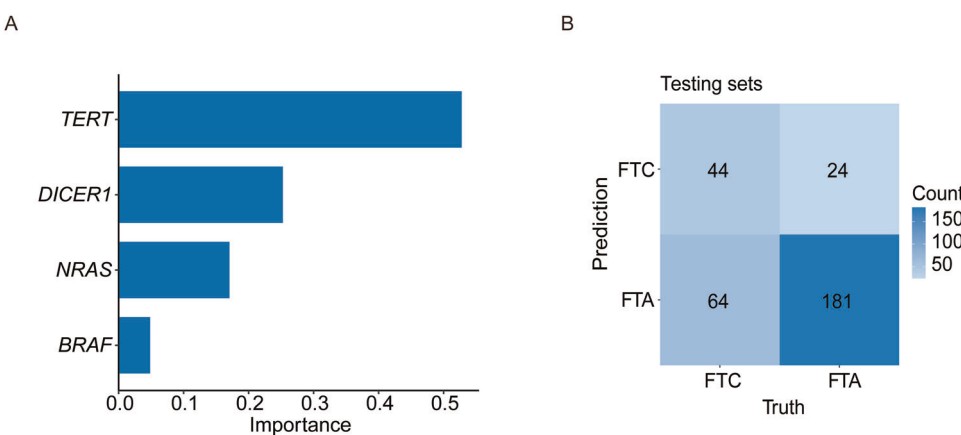

**Figure EV1. Features and evaluation result of gene-based model.**

(A) The importance rank of four selected features. (B) Confusion matrix of the four-gene model.

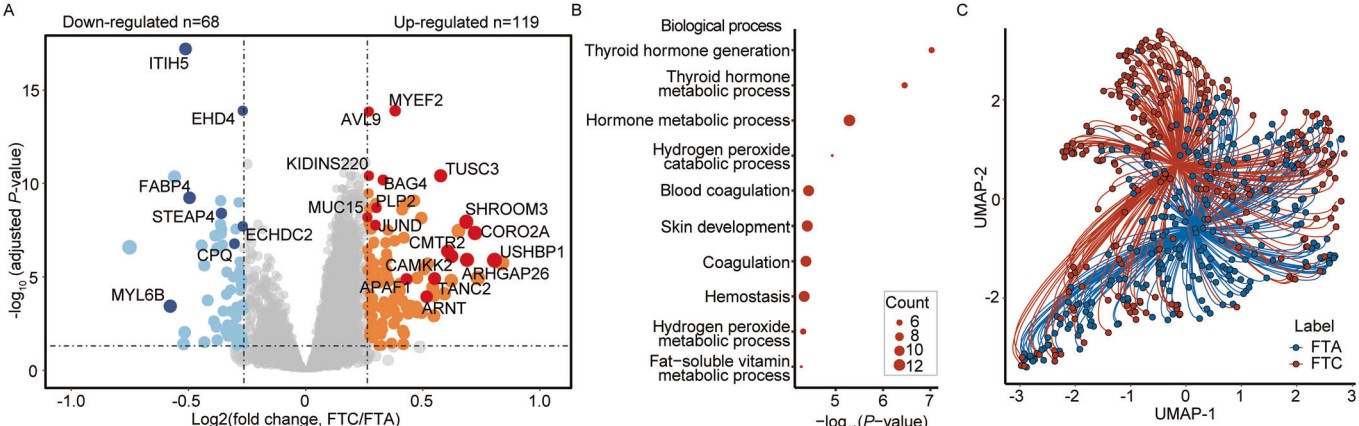

**Figure EV2. Comparative proteomic analysis for FTC vs. FTA in TMT discovery dataset.**

(A) Differentially expressed proteins (DEPs) in FTC ($n = 224$) and FTA ($n = 261$). Thresholds of significantly dysregulated proteins: fold change >1.2 with Benjamini & Hochberg adjusted $P < 0.05$ (Welch's *t*-test). The highlighted proteins are the features selected by the model in Fig. EV3. (B) Gene ontology (GO) biological process enrichment. X-axis represents the $P$ and node size indicates the protein count of enriched items. $P$ values are calculated using a one-sided Fisher's Exact Test. (C) UMAP plot showing FA and FTC are partially resolved using 187 DEPs.

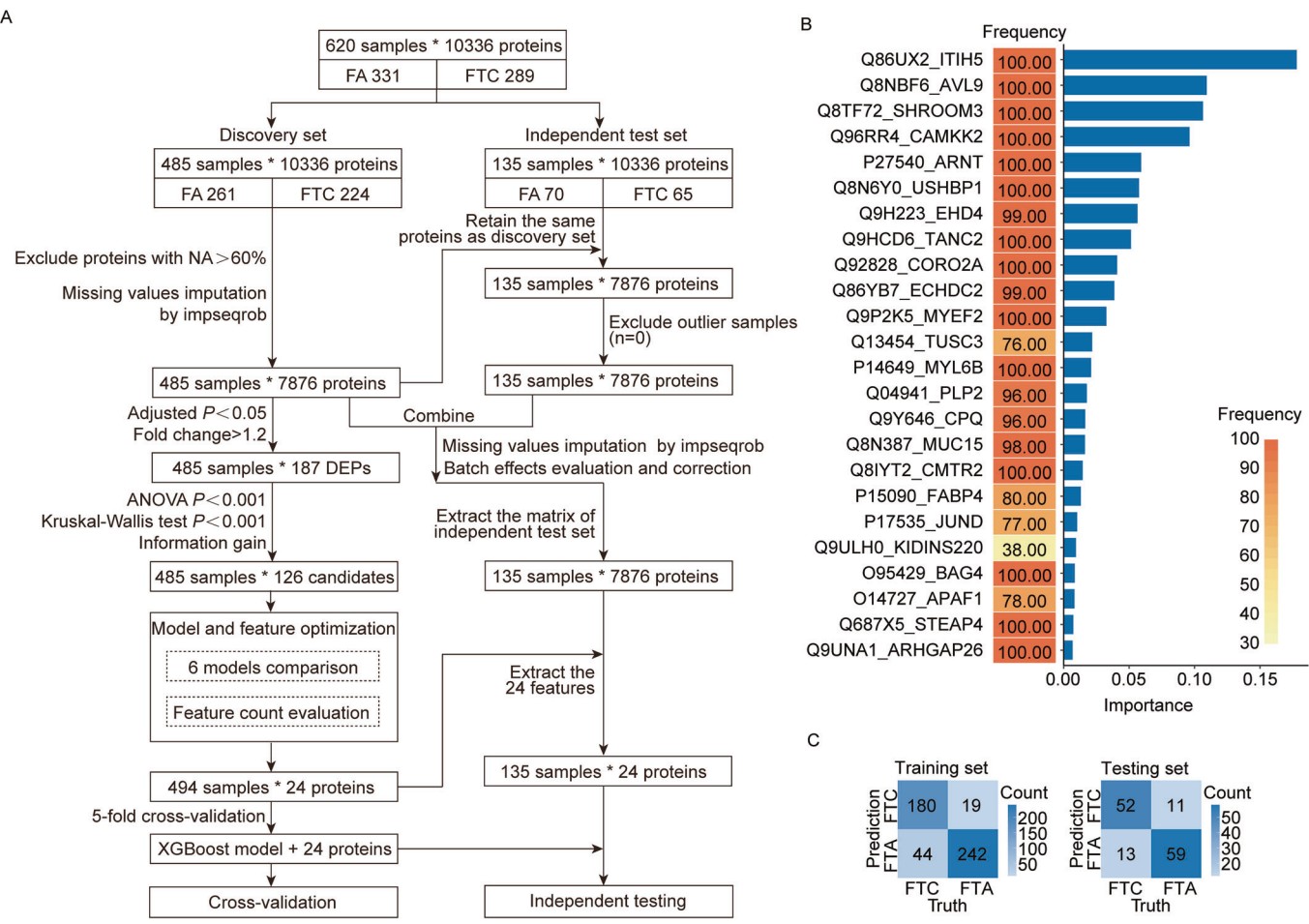

**Figure EV3. Modeling using the discovery proteomic data.**

(A) Schematic of XGBoost model construction. Samples firstly are divided into a discovery set and an independent set. Feature selection and model training are based on the discovery set and model performance evaluation is based on the independent test set. (B) Importance ranking of selected protein features and the frequency of 24 feature proteins when conducting 100 times feature selection on the training set. The orange color bar shows the selection frequency. (C) Confusion matrix of the 24-protein classifier. The blue color bar indicates the sample counts.

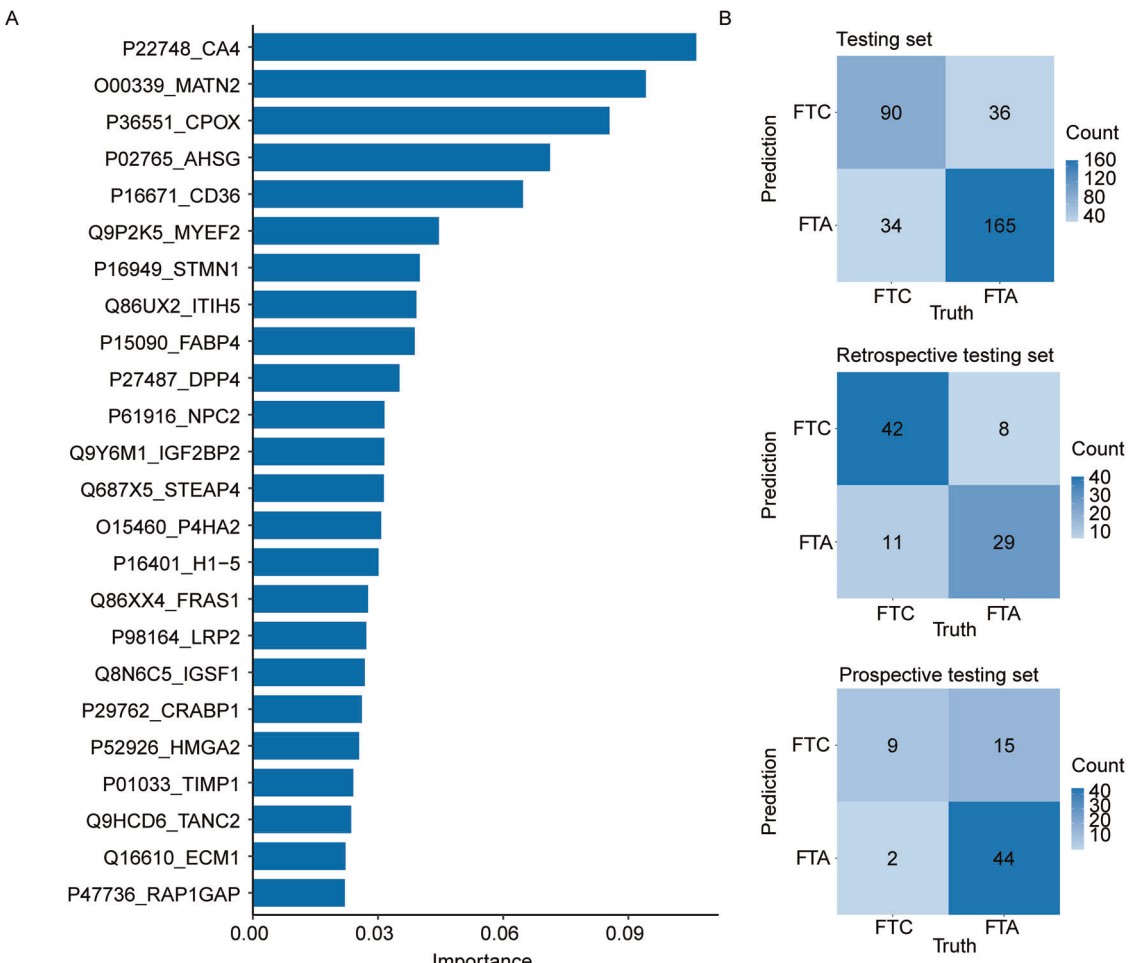

**Figure EV4. Targeted proteomic data-based feature importance and model performance.**

(A) The 24 selected targeted protein features and their importance rankings. (B) Confusion matrix of the 24-protein classifier. Colors indicate the sample counts.

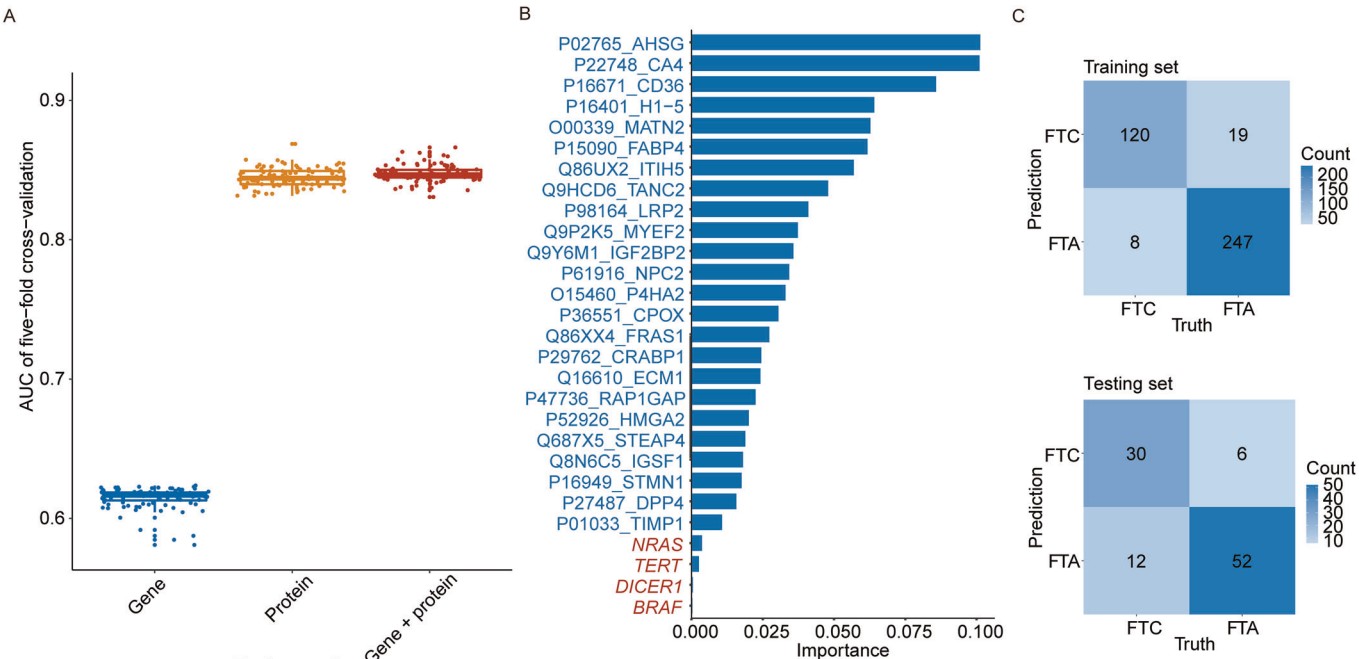

**Figure EV5. Model performance comparison and characters of the combined feature-based model.**

(A) The area under the curve (AUC) comparison of gene-, protein-, gene and protein-based models of five-fold cross-validation for one-hundred-time iterations. Each dot indicates one iteration. The box plots ($n = 100$ per group) are defined as follows: the lower and upper edges of the box represent the 25th percentile (first quartile) and the 75th percentile (third quartile), respectively; the central line represents the 50th percentile (median); and the whiskers span from the 0th percentile (minimum) to the 100th percentile (maximum). (B) The importance ranking of features. Protein features are colored in blue and gene features are colored in red. (C) Confusion matrix of the combined feature classifier. Colors indicate the sample counts.

