## [Peer Review File · EMBO Molecular Medicine]

A protein-based classifier for differentiating follicular thyroid adenoma and carcinoma

Yaoting Sun, He Wang, Lu Li, Jianbiao Wang, Wanyuan Chen, Li Peng, Pingping Hu, Jing Yu, Xue Cai, Nan Yao, Yan Zhou, Jiatong Wang, Yingrui Wang, Liqin Qian, Weigang Ge, Mengni Chen, Feng Yang, Zhiqiang Gui, Wei Sun, Zhihong Wang, Minghua Ge, Yi He, Guangzhi Wang, Yongfu Zhao, Huanjie Chen, Xiaohong Wu, Yuxin Du, Wenjun Wei, Fan Wu, Dingcun Luo, Xiangfeng Lin, Haitao Zheng, Xin Zhu, Bei Wei, Jiafei Shen, Jincao Yao, Zhennan Yuan, Tong Liu, Jun Pan, Yifeng Zhang, Yang-Fan Lv, Qiao-Nan Guo, Qi-Jun Wu, Ting-Ting Gong, Ting Chen, Shu Zheng, Jingqiang Zhu, Hanqing Liu, Chuang Chen, Hong Han, Sathiyamoorthy Selvarajan, Michael Mingzhao Xing, Kennichi Kakudo, Erik Alexander, Yijun Wu, Yu Wang, Dong Xu, Hao Zhang, Xiu Nie, Oi Lian Kon, N. Iyer, Zhiyan Liu, Yi Zhu, Haixia Guan, and Tiannan Guo

Corresponding authors: Tiannan Guo (guotiannan@westlake.edu.cn) , Yi Zhu (zhuyi@westlake.edu.cn), Haixia Guan (guanhaixia@gdph.org.cn)

Review Timeline:

Submission Date:	13th Jan 25
Editorial Decision:	31st Mar 25
Revision Received:	15th Apr 25
Editorial Decision:	21st Apr 25
Revision Received:	22nd Apr 25
Accepted:	25th Apr 25

Editor: Jingyi Hou

Transaction Report:

31st Mar 2025

Dear Tiannan,

Thank you again for submitting your work to EMBO Molecular Medicine. I apologize for the significant delay in the review process, which was due to the late arrival of the referees' reports. We have now received feedback from the two referees who agreed to evaluate your manuscript. As you will see in the reports below, the referees are very supportive and have raised only minor issues that should be addressed in a minor revision of the manuscript.

The referees' recommendations are clear, so I won't repeat the points listed below. All of the raised issues need to be addressed. We would welcome the submission of a revised version within three months for further consideration. As you may already know, our editorial policy allows in principle a single round of major revision, and it is therefore essential to provide responses to the referees' comments that are as complete as possible.

Please also contact us as soon as possible if similar work is published elsewhere. If other work is published, we may not be able to extend the revision period beyond three months.

I look forward to receiving your revised manuscript.

Yours sincerely,
Jingyi

Jingyi Hou
Senior Editor
EMBO Molecular Medicine

We require:

- 1) A .docx formatted version of the manuscript text (including legends for main figures, EV figures and tables). Please make sure that the changes are highlighted to be clearly visible.
- 2) Individual production quality figure files as .eps, .tif, .jpg (one file per figure). For guidance, download the 'Figure Guide PDF': (<https://www.embopress.org/page/journal/17574684/authorguide#figureformat>).
- 3) A .docx formatted letter INCLUDING the reviewers' reports and your detailed point-by-point responses to their comments. As part of the EMBO Press transparent editorial process, the point-by-point response is part of the Review Process File (RPF), which will be published alongside your paper.
- 4) A complete author checklist, which you can download from our author guidelines (<https://www.embopress.org/page/journal/17574684/authorguide#submissionofrevisions>). Please insert information in the checklist that is also reflected in the manuscript. The completed author checklist will also be part of the RPF.
- 5) Please note that all corresponding authors are required to supply an ORCID ID for their name upon submission of a revised manuscript.

6) It is mandatory to include a 'Data Availability' section after the Materials and Methods. Before submitting your revision, primary datasets produced in this study need to be deposited in an appropriate public database, and the accession numbers and database listed under 'Data Availability'. Please remember to provide a reviewer password if the datasets are not yet public (see <https://www.embopress.org/page/journal/17574684/authorguide#dataavailability>).

12) Author contributions: You will be asked to provide CRediT (Contributor Role Taxonomy) terms in the submission system. These replace a narrative author contribution section in the manuscript.

13) A Conflict of Interest statement should be provided in the main text.

14) Every published paper now includes a 'Synopsis' to further enhance discoverability. Synopses are displayed on the journal webpage and are freely accessible to all readers. They include a short stand first (maximum of 300 characters, including space) as well as 2-5 one-sentences bullet points that summarizes the paper. Please write the bullet points to summarize the key NEW

findings. They should be designed to be complementary to the abstract - i.e. not repeat the same text. We encourage inclusion of key acronyms and quantitative information (maximum of 30 words / bullet point). Please use the passive voice. Please attach these in a separate file or send them by email, we will incorporate them accordingly.

Please also suggest a visual abstract to illustrate your article as a PNG file 550 px wide x 300-600 px high.

15) All Materials and Methods need to be described in the main text using our 'Structured Methods' format. According to this format, the Methods section includes a Reagents and Tools Table (listing key reagents, experimental models, software and relevant equipment and including their sources and relevant identifiers) followed by a Methods and Protocols section describing the methods, ideally using a step-by-step protocol format. The aim is to facilitate adoption of the methodologies across labs.

Please download and fill our Reagents and Tools Table template (.docx), which you can find in our author guidelines: <https://www.embopress.org/page/journal/17574684/authorguide#structuredmethods>

When submitting your revised manuscript, please DO NOT include the Reagents and Tools Table in the Methods section of the manuscript but upload it as a separate file choosing the file type "Reagent Table".

***** Reviewer's comments *****

Referee #1 (Comments on Novelty/Model System for Author):

Please see comments to the authors

Referee #1 (Remarks for Author):

Here are my Comments on "A protein-based classifier for differentiating follicular thyroid adenoma and carcinoma" by Sun et al.

This is an overall well-conducted and thoughtfully presented study. The methodology appears sound, and the conclusions are generally well supported by the data. The authors have also taken care to address several potential limitations of their work, which is commendable. However, I have several minor points and questions that I would like the authors to consider:

Line 170: The reported female-to-male ratio of 2.4:1 raises the question of whether this reflects the known gender distribution in the general population for this condition. It would be useful for the authors to clarify if this ratio is representative.

Line 441 (Figure 2c) and Sup. Figure 8: The inclusion of statistical tests on the AUC values appears unnecessary. These tests do not add meaningful information and, more importantly, the underlying AUC distributions are unlikely to meet the assumptions required for Welch's test. I suggest removing these tests or providing justification for their use.

Line 573: Missing values were imputed with zeros prior to PCA. This approach makes the resulting PCA heavily dependent on the rate of missing values in each sample. The authors acknowledge this issue in the text, which leads to the question: why not filter samples based on missing value rates in a more consistent and principled manner from the outset? Such an analysis could strengthen the conclusions.

Lines 479-581: The imputation approach used may still lead to data leakage, as the test data are potentially informed by statistics derived from the training set. The authors should clarify how this risk was mitigated, or consider an alternative approach that fully isolates the test data.

Lines 582-587: While Combat was used for batch correction, it is known that this method can affect correlations between variables. It would be valuable to assess how within-batch protein correlations are impacted by Combat, both before and after correction.

Lines 642-643: In the quality control step, PCA was performed using a centred but unscaled discovery set, with missing values imputed as zero. This decision, combined with the zero imputation, likely increases the influence of missing data on the PCA results. Would it not be better to scale the data?

Line 794, Sup. Fig. 2: It would enhance the study to include a UMAP plot coloured by centre. Additionally, showing PCA plots before and after batch correction would help to visualize the impact of the correction procedure.

In summary, this is a valuable contribution to the field, and addressing the points above will further strengthen the clarity and rigor of the study.

Referee #3 (Comments on Novelty/Model System for Author):

Sun et al. provide a manuscript in which they aim to find a molecular classifier for differentiating follicular thyroid adenoma and carcinoma. The medical reasons for this goal are substantiated but the approach is not new. Nevertheless, I admit that a construction of the research model is original and has chances to find an improved classifier than the previously published ones. The Authors first analyzed the utility of DNA based data and then they decided to construct a protein-based classifier. They analyzed also whether a combination of DNA data and protein mass spectrometry data would result in better accuracy, with negative response. Their approach is sound and important from the clinical point of view. The proposed protein-based classifier may be applied in the routine diagnostics

Referee #3 (Remarks for Author):

I approve the manuscript as it is

Point-by-point response to the reviewers' comments

We thank all reviewers for their thorough and positive evaluation of our manuscript entitled "*A protein-based classifier for differentiating follicular thyroid adenoma and carcinoma*" [EMM-2025-21197]. The reviewer's positive assessment of our methodology and conclusions is greatly encouraging. We believe that in the revised version, all points are now addressed, further strengthening our manuscript. In a nutshell, we included some additional analyses to solve the concerns from Reviewer #1 regarding data preprocessing. In the pages below, each of the reviewers' comments are addressed in more detail. All the changes are highlighted to be clearly visible. We provide data directly in those cases where it wasn't possible to incorporate it into the revised manuscript.

***** Reviewer's comments *****

Referee #1 (Remarks for Author):

Here are my Comments on "A protein-based classifier for differentiating follicular thyroid adenoma and carcinoma" by Sun et al.

This is an overall well-conducted and thoughtfully presented study. The methodology appears sound, and the conclusions are generally well supported by the data. The authors have also taken care to address several potential limitations of their work, which is commendable. However, I have several minor points and questions that I would like the authors to consider:

Reply: We sincerely thank the reviewer for the thorough evaluation and valuable comments on our manuscript. After carefully considered all the points and questions raised, we made some adjustments in the revised manuscript. We appreciate your questions and hope that this clarification adequately addresses your concerns.

1. Line 170: The reported female-to-male ratio of 2.4:1 raises the question of whether this reflects the known gender distribution in the general population for this condition. It would be useful for the authors to clarify if this ratio is representative.

Reply: Thank you for your comment. Our reported ratio of 2.4:1 accurately reflects the

natural gender distribution of this condition, supporting the representativeness of our study population and the generalizability of our findings.

Our data are also in line with the results of published population cohort studies(Aschebrook-Kilfoy *et al*, 2013; Shan *et al*, 2025), which consistently illustrate a female predominance in follicular thyroid neoplasms, with ratios typically ranging from 2:1 to 3:1. This gender disparity is well-documented in the literatures and supports the representativeness of our study population.

2. Line 441 (Figure 2c) and Sup. Figure 8: The inclusion of statistical tests on the AUC values appears unnecessary. These tests do not add meaningful information and, more importantly, the underlying AUC distributions are unlikely to meet the assumptions required for Welch's test. I suggest removing these tests or providing justification for their use.

Reply: Thanks for your constructive suggestion. We have removed these tests in the figures, as shown below:

Fig. 2C

Fig. EV5A

3. Line 573: Missing values were imputed with zeros prior to PCA. This approach makes the resulting PCA heavily dependent on the rate of missing values in each sample. The authors acknowledge this issue in the text, which leads to the question: why not filter samples based on missing value rates in a more consistent and principled manner from the outset? Such an analysis could strengthen the conclusions.

Reply: Thank you for your comments. We agree your comment that "the resulting PCA heavily dependent on the rate of missing values in each sample". And we indeed found the outliers detected by our PCA have a high missing value rates (ranging from 30.4% to 36.3%, much higher than other samples). But in multi-plex TMT proteomics, the samples within each batch share the same missing value rate, which means filtering solely on missing rates would eliminate entire batches. Therefore, we adopted a more conservative way (PCA, which considers the missing value rates and the protein expression together) to locate and remove the outlying samples. To validate the consistency of this step, we also tried PCA using different imputation algorithms and detected similar set of outliers, as shown below:

Additionally, in our preprocessing, we not only removed the outliers, but also filtered unreliable proteins with missing value rates $> 60\%$, which further reduces the sample missing value rates (range 5.7% - 18.0% with mean = 9.3% and median = 9.1%, after removing proteins with high missing value rates). Then we imputed the missing values using more powerful and robust algorithm *impseqrob* selected by *NAGuideR*(Wang et

al, 2020). Therefore, our preprocessing pipeline has ensured the removal of outlier samples (including those with high missing value rates) and the robustness of further analyses.

4. Lines 479-581: The imputation approach used may still lead to data leakage, as the test data are potentially informed by statistics derived from the training set. The authors should clarify how this risk was mitigated, or consider an alternative approach that fully isolates the test data.

Reply: Thank you for raising this important methodological concern regarding potential data leakage during imputation. We carefully designed our preprocessing procedure to prevent data leakage from test set to training set. Our imputation strategy follows the steps:

Step 1. We first performed imputation on the training set in isolation, using only information contained within the training data.

Step 2. We combined the imputed training set and test set together and ran imputation algorithm `impseqrob`. In this process, the information derived from the already-imputed training set were utilized to predict missing values in the test set. In the meantime, the training set will not be influenced by the test data.

These steps follows the instructions from Sean Whalen's paper (**Figure 1d**)(Whalen *et al*, 2022), which described how to avoid the data leakage in preprocessing steps.

In fact, we should ensure that no information from the test set influence the training process to avoid overestimate the performance of the models, but the information from training set can be utilized to further improve the curation of test set, which is an effective practice in machine learning modelling.

Additionally, we imputed test data in an isolated way, validated our model using the newly imputed test data, and found the performance is still satisfying, although slightly lower than our previous results:

Performance on Test Set	Original Results	Isolated Imputation
AUC	0.899 (0.849-0.949)	0.887 (0.833-0.942)
Accuracy	0.822 (0.748-0.878)	0.807 (0.732-0.865)
Sensitivity	0.800 (0.685-0.880)	0.785 (0.669-0.868)
Specificity	0.843 (0.738-0.911)	0.829 (0.722-0.900)
PPV	0.825 (0.711-0.901)	0.810 (0.694-0.888)
NPV	0.819 (0.713-0.892)	0.806 (0.698-0.881)

In summary, our imputation strategy is robust and avoids the risk of data leakage.

5. Lines 582-587: While Combat was used for batch correction, it is known that this method can affect correlations between variables. It would be valuable to assess how within-batch protein correlations are impacted by Combat, both before and after correction.

Reply: Thank you for your suggestion. According to your instruction, we conducted additional analysis by calculating Pearson correlations between proteins within each batch. Our results demonstrate that the correlation patterns remain highly consistent before and after correction (according to the boxplots shown below), confirming that our approach preserves the underlying biological relationships while effectively addressing technical variation. Notably, batch 38 exhibits a higher IQR for protein correlations, likely due to the fact that only four samples remain after quality control, which contributes to the divergence in protein correlations.

6. Lines 642-643: In the quality control step, PCA was performed using a centred but unscaled discovery set, with missing values imputed as zero. This decision, combined with the zero imputation, likely increases the influence of missing data on the PCA results. Would it not be better to scale the data?

Reply: Thank you for your comments. We have justified the choice of imputation before PCA in the above second point. As for scaling, we directly used the expression ratio matrix from Proteome Discoverer software in our analyses, and these ratios represent how many folds of the abundances in analyzed samples when compared with the abundances in pooled samples, which have already been scaled. Therefore, the expression ratio distributions of all proteins have similar ranges and scales, making the scaling in PCA (to avoid features with larger variances dominating the principal components) unnecessary.

7. Line 794, Sup. Fig. 2: It would enhance the study to include a UMAP plot coloured by centre. Additionally, showing PCA plots before and after batch correction would help to visualize the impact of the correction procedure.

Reply: Thanks for pointing out this. We have added the UMAP plot colored by center to the manuscript (Appendix Figure S1D), as shown below:

As for the PCA plots before and after batch correction, instead of PCA, we did UMAP to evaluate the batch effects since PCA is more focusing on global structures while UMAP is better in finding local structures (Yang *et al*, 2021), which aligns with the situation of batch effects in our datasets (only a few batches show obvious batch effects, e.g., top left corner of left figure below: batch 35-38). After correction, the samples from different batches are well-mixed in the UMAP visualization (Right figure below), showing no batch effects. The UMAP figures have been added to the manuscript (Appendix Figure S1B and 1C).

(B) UMAP analysis before and (C) after batch correction.

In summary, this is a valuable contribution to the field, and addressing the points above will further strengthen the clarity and rigor of the study.

Reply: We thank the reviewer again for the valuable suggestions and feedback.

Referee #3 (Comments on Novelty/Model System for Author):

Sun et al. provide a manuscript in which they aim to find a molecular classifier for differentiating follicular thyroid adenoma and carcinoma. The medical reasons for this goal are substantiated but the approach is not new. Nevertheless, I admit that a construction of the research model is original and has chances to find an improved classifier than the previously published ones. The Authors first analyzed the utility of DNA based data and then they decided to construct a protein-based classifier. They analyzed also whether a combination of DNA data and protein mass spectrometry data would result in better accuracy, with negative response. Their approach is sound and important from the clinical point of view. The proposed protein-based classifier may be applied in the routine diagnostics

Referee #3 (Remarks for Author):

I approve the manuscript as it is

Reply: We sincerely thank Referee #3 for the careful and insightful review of our manuscript. We greatly appreciate the recognition of the originality and clinical relevance of our research model, particularly the acknowledgment of our efforts in constructing a protein-based classifier for differentiating follicular thyroid adenoma and carcinoma. Your positive evaluation of our methodological approach and its potential application in routine diagnostics is highly encouraging.

References

Aschebrook-Kilfoy B, Grogan RH, Ward MH, Kaplan E, Devesa SS (2013) Follicular thyroid cancer incidence patterns in the United States, 1980-2009. *Thyroid* 23: 1015-1021

Shan R, Li X, Chen J, Chen Z, Cheng YJ, Han B, Hu RZ, Huang JP, Kong GL, Liu H *et al* (2025) Interpretable Machine Learning to Predict the Malignancy Risk of Follicular Thyroid

Neoplasms in Extremely Unbalanced Data: Retrospective Cohort Study and Literature Review.

JMIR Cancer 11: e66269

Wang S, Li W, Hu L, Cheng J, Yang H, Liu Y (2020) NAGuideR: performing and prioritizing missing value imputations for consistent bottom-up proteomic analyses. *Nucleic Acids Res* 48: e83

Whalen S, Schreiber J, Noble WS, Pollard KS (2022) Navigating the pitfalls of applying machine learning in genomics. *Nat Rev Genet* 23: 169-181

Yang Y, Sun H, Zhang Y, Zhang T, Gong J, Wei Y, Duan YG, Shu M, Yang Y, Wu D *et al* (2021) Dimensionality reduction by UMAP reinforces sample heterogeneity analysis in bulk transcriptomic data. *Cell Rep* 36: 109442

21st Apr 2025

Dear Tiannan,

Thank you for sending us your revised manuscript. We are overall satisfied with the revisions made. Before we can formally accept the manuscript for publication, a few remaining editorial level issues still need to be addressed:

1. Since the final manuscript contains only two figures, we will change the manuscript type to "Report".
2. Please remove the figures from the manuscript file.
3. Authors
 - There is a name discrepancy that needs to be resolved- Mingzhao Xing in the manuscript file vs. Michael Mingzhao Xing in the submission system.
 - Please remove the list of consortia names on page 30.
4. Please ensure that the funding information listed in the submission system and the manuscript is consistent. Currently, 2024ZD0525603 is missing from the submission system, and National Natural Science Foundation of China U21A20427 is missing from the manuscript file. Please resolve this discrepancy.
5. "Conflicts of interest" needs to be renamed to "Disclosure and Competing Interests Statement".
6. The code availability should be part of the Data availability section, so no additional heading is needed. Remove the paragraph beginning with "All the data will be made publicly available upon publication of this manuscript...".
7. Remove the Authors' contribution section from the manuscript file.
8. "The paper explained" should be included in the main manuscript file.
9. A callout for Appendix Table S2 is missing, please fix it.
10. Appendix: Please add a title page with a Table of Content and page numbers to show where each Appendix item is located.
11. I have slightly modified the synopsis text(see attached). Please let me know if this is acceptable as is or if you would prefer to make additional modifications.
12. Please address the following issues with figure legends
 - Please note that the box plots need to be defined in terms of minima, maxima and percentile in the legends of figures 2C, EV5A
 - Please note that information related to n is missing in the legend of figure EV5 A.
13. The manuscript sections should be in the following order: Title page - Abstract & Keywords - Introduction - Results - Discussion -Methods - Data Availability - Acknowledgments - DisclosureStatement & Competing Interests - References - Figure Legends -(Main Tables with legends if applicable) - Expanded View FigureLegends.

Please submit your revised manuscript within two weeks. I look forward to seeing a revised form of your manuscript as soon as possible.

Kind regards,
Jingyi

Jingyi Hou
Senior Editor
EMBO Molecular Medicine

*** Instructions to submit your revised manuscript ***

- 1) a .docx formatted version of the manuscript text (including Figure legends and tables)
 - 2) Separate figure files*
 - 3) supplemental information as Expanded View and/or Appendix. Please carefully check the authors guidelines for formatting Expanded view and Appendix figures and tables at <https://www.embopress.org/page/journal/17574684/authorguide#expandedview>
 - 4) a letter INCLUDING the reviewer's reports and your detailed responses to their comments (as Word file).
 - 5) The paper explained: EMBO Molecular Medicine articles are accompanied by a summary of the articles to emphasize the major findings in the paper and their medical implications for the non-specialist reader. Please provide a draft summary of your article highlighting
 - the medical issue you are addressing,
 - the results obtained and
 - their clinical impact.This may be edited to ensure that readers understand the significance and context of the research. Please refer to any of our published articles for an example.
 - 6) Author contributions: the contribution of every author must be detailed in a separate section.
 - 7) EMBO Molecular Medicine now requires a complete author checklist (<https://www.embopress.org/page/journal/17574684/authorguide>) to be submitted with all revised manuscripts. Please use the checklist as guideline for the sort of information we need WITHIN the manuscript. The checklist should only be filled with page numbers where the information can be found. This is particularly important for animal reporting, antibody dilutions (missing) and exact values and n that should be indicated instead of a range.
 - 8) Every published paper now includes a 'Synopsis' to further enhance discoverability. Synopses are displayed on the journal webpage and are freely accessible to all readers. They include a short stand first (maximum of 300 characters, including space) as well as 2-5 one sentence bullet points that summarise the paper. Please write the bullet points to summarise the key NEW findings. They should be designed to be complementary to the abstract - i.e. not repeat the same text. We encourage inclusion of key acronyms and quantitative information (maximum of 30 words / bullet point). Please use the passive voice. Please attach these in a separate file or send them by email, we will incorporate them accordingly.
- You are also welcome to suggest a striking image or visual abstract to illustrate your article. If you do please provide a jpeg file 550 px-wide x 300-600px high.
- 9) A Conflict of Interest statement should be provided in the main text
 - 10) Please note that we now mandate that all corresponding authors list an ORCID digital identifier. This takes <90 seconds to complete. We encourage all authors to supply an ORCID identifier, which will be linked to their name for unambiguous name identification.

Currently, our records indicate that the ORCID for your account is 0000-0003-3869-7651.

Please click the link below to modify this ORCID:
Link Not Available

11) Include a Reagents and Tools Table as part of the Methods section, which can be downloaded from our author guidelines (<https://www.embopress.org/page/journal/17574684/authorguide#structuredmethods>)

Photos 400-800 DPI

*Additional important information regarding figures and illustrations can be found at <https://bit.ly/EMBOPressFigurePreparationGuideline>. See also figure legend preparation guidelines: <https://www.embopress.org/page/journal/17574684/authorguide#figureformat>

***** Reviewer's comments *****

The authors addressed the remaining editorial issues.

25th Apr 2025

Dear Tiannan,

Thank you for sending us the revised manuscript. We are pleased to inform you that your manuscript is accepted for publication and is now being sent to our publisher to be included in the next available issue of EMBO Molecular Medicine.

Kind regards,
Jingyi

Jingyi Hou
Senior Editor
EMBO Molecular Medicine
